# Causal attributions shape the formation of novel ability self-beliefs
Annalina V. Mayer [1,4] ✉, Alexander Schröder[1,4], David S. Stolz[1], Nora Czekalla [1], Frieder M. Paulus [1], Sören Krach [1] ✉, Tobias Kube [2,3,5] & Laura Müller-Pinzler[1,5]

Healthy individuals typically attribute success to internal and failure to external factors, while people with depression and low self-esteem tend to do the opposite. At the same time, depression and low self-esteem are associated with negatively biased self-related learning and self-beliefs. Here, we used a validated self-related learning task to investigate how internal versus external attributions of performance feedback affect the formation of self-beliefs about ability and how these processes relate to depressive symptoms and self-esteem. Drawing on a computational model that incorporates prediction error valence and causal attributions, we found that participants updated their ability beliefs less when feedback was attributed to external causes. Furthermore, individuals with higher levels of depression and lower self-esteem showed a stronger negativity bias in learning and lower self-esteem was linked to a reduced self-serving attributional bias. These findings provide insight into cognitive mechanisms associated with negative self-related ability beliefs and may help to inform our understanding of processes linked to depressive symptoms.

Attributional styles, referred to as individual tendencies to explain the causes of events, play a crucial role in shaping people's beliefs and their understanding of themselves and the world. In particular, attributional styles reflect whether individuals interpret life events as resulting from internal or external, stable or unstable, and global or specific causes[1]. A person with a pessimistic attributional style tends to view negative events as internally caused, stable, and global. Conversely, an optimistic attributional style tends to foster resilience by attributing setbacks to external, unstable, and situation-specific factors[2]. Understanding attributional styles thus offers critical insight into cognitive vulnerability and resilience across various aspects of psychological functioning.

Attributional styles have been linked to mental health outcomes[3]. For instance, while healthy individuals tend to attribute successes to internal causes (e.g., their abilities) and failures to external causes (e.g., bad luck)[4], a pattern commonly referred to as a self-serving attributional bias or self-serving attributional style, individuals with depression often show the opposite pattern[2,5,6]. In fact, a pessimistic attributional style is associated with low self-esteem[7–9] and is considered a key cognitive factor in the development and maintenance of depression and other mental disorders[1,10–13].

Depression and low self-esteem are not only linked to a more pessimistic attributional style, but also to a tendency to learn about oneself in an overly negative way. For example, individuals with major depressive disorder show difficulties in updating their negative beliefs after unexpected positive experiences[14,15]. Individuals with clinical depression and those with elevated depression levels were also found to interpret ambiguous situations more negatively and less positively, especially when they contain self-relevant information[16]. Similarly, individuals with low self-esteem tend to learn less from social feedback than those with high self-esteem and are more likely to expect others to dislike them[17]. Further, within self-referential learning, healthy individuals show a negativity bias when updating beliefs about their own characteristics and abilities[18,19], which is exacerbated in individuals with low self-esteem[18,20] and clinical depression[21]. A separate line of work shows that healthy individuals tend to update their expectations about future events in an optimistically biased way[22]. In contrast, individuals with depression show a reduced or absent optimistic update bias[23,24], a pattern that has also been observed in the context of social evaluation[25].

Biased self-beliefs in depression and individuals with low self-esteem might be related to differences in attributional styles, since causal attributions may influence how beliefs are formed and updated. To illustrate this, consider the following example: A good student receives a poor grade on a math exam. How the student interprets this outcome, whether as a result of internal or external causes, shapes how they think about their mathematical ability. If the student believes the low grade was due to their lack of understanding of the material (internal attribution), they are more likely to lower their belief about their ability. On the other hand, if the student attributes the poor grade to the exam being unfair or the teacher being biased

[1]University of Luebeck, Department of Psychiatry and Psychotherapy, Lübeck, Germany. [2]Goethe University Frankfurt, Frankfurt am Main, Germany. [3]Rheinland-Pfälzische Technische Universität Kaiserslautern-Landau, Kaiserslautern, Germany. [4]These authors contributed equally: Annalina V. Mayer, Alexander Schröder. [5]These authors jointly supervised this work: Tobias Kube, Laura Müller-Pinzler. ✉e-mail: ann.mayer@uni-luebeck.de; soeren.krach@uni-luebeck.de

(external attribution), they may dismiss the feedback and continue to hold a firm belief about their ability. If the student has the tendency to attribute failures to internal causes, these failures are more likely to shape their self-beliefs than successes, resulting in negatively biased beliefs.

A number of behavioral studies have demonstrated this relationship of causal attributions and subsequent self-beliefs in a performance context: when participants attributed a successful performance to internal causes (i.e., their ability), this heightened their self-efficacy, while attributions to external causes (i.e., good luck) lowered their self-efficacy[26,27]. More recent advancements in neuroscience and psychology have applied computational modeling techniques to explore how causal attributions shape beliefs about action-outcome relationships in a reinforcement-learning task[28,29]. Participants gave less weight to outcomes they believed were caused by external factors—the interference of a hidden agent—rather than their own actions. In other words, when participants thought an outcome was manipulated, they learned less from it. This process resulted in biased learning depending on whether the hidden agent changed the outcome positively or negatively[28,29].

Taken together, this body of work highlights the importance of causal attributions in shaping belief formation and updating. However, it also leaves two important gaps. First, most prior research has conceptualized attributional style primarily as a stable interindividual trait[30], rather than examining how causal attributions made in real time influence belief updating as it unfolds. Second, although recent computational approaches have incorporated trial-by-trial attributions[28,29,31], their focus has largely been on learning about action–outcome contingencies in the external environment. Considerably less attention has been paid to how these attributional processes shape learning about one's own abilities. Addressing these gaps is essential for understanding how moment-to-moment causal interpretations contribute to the formation of self-related ability beliefs[32].

In this study, we thus aim to answer the following questions: First, do momentary attributions of failures and successes generally influence self-related learning and consequently shape self-related ability beliefs? Second, are depressive symptoms and self-esteem related to different attributional patterns during self-related learning, and does this contribute to biased ability beliefs?

Using a validated learning task[18,20,21,32,33], we investigated how internal vs. external attributions of performance feedback shape self-related learning in novel ability domains. Drawing on computational learning models that account for prediction error valence[18,20,21,33] and that have been further extended to consider causal attributions, we first hypothesized that participants would learn less from feedback attributed to external causes compared to feedback attributed to their own abilities. Second, we hypothesized that depressive symptom severity and self-esteem would be associated with biases in learning and attributions. Specifically, based on research pointing to deficits in learning from novel positive experiences in people with elevated levels of depression[15,24,34], we expected that participants with higher levels of depression would show a stronger negative bias in how they learned from the feedback. Similarly, we expected that lower self-esteem would be linked to a stronger negative bias, as shown before[18,20]. As implicated by the attributional theory of depression[1,10–13], we also anticipated that individuals with more severe depressive symptoms and lower self-esteem would be more likely to attribute failures (worse-than-expected feedback) to internal causes and successes (better-than-expected feedback) to external causes, reflecting a reduced or even reversed self-serving attributional bias.

## Methods
### Participants
The sample consisted of 66 university students (50 female, 16 male) aged between 19 and 33 years ($M = 22.8$, $SD = 3.3$) who were recruited from the University of Kaiserslautern-Landau, Germany. All participants were fluent in German, had either normal or corrected-to-normal vision, and gave written informed consent. The study was approved by the Ethics Committee of the University of Kaiserslautern-Landau (reference number: LEK517add) and conducted in accordance with the ethical guidelines of the American Psychological Association. Participants received 10€ or partial course credit upon completion of the study, as approved by the department's research participation system. Two participants were excluded due to insufficient response variability, as indicated by a lack of cursor movement, suggesting low task engagement. This left a total of 64 participants to be analyzed, of which 49 female and 15 were male. Gender was determined based on information provided by the participants.

### Learning of own performance task
Participants completed an adapted version of a validated self-related learning task, the learning of own performance (LOOP) task. The LOOP task enables participants to incrementally learn about their alleged ability in estimating properties, for example, the height of houses or the weight of animals. The original version of the LOOP task was introduced and validated in a series of behavioral and neuroimaging studies[18,20,21,33,35]. In this study, we used an adapted version including a condition that allowed participants to attribute performance feedback to an intervention by the computer (or "agent") rather than their own ability.

On a trial-by-trial basis, participants were asked to estimate different properties and received predefined performance feedback in four distinct estimation categories (Fig. 1A). As part of the cover story, the LOOP task was presented as a study on cognitive estimation ability, with feedback described as reflecting participants' performance. In reality, all feedback was pre-programmed and independent of actual responses in the estimation task. Two categories were randomly assigned to predominantly better-than-expected feedback (High Ability condition), and two to predominantly worse-than-expected feedback (Low Ability condition). Participants were informed that in two categories (Agent condition), the computer might alter the feedback, such that performance could be either improved in one category or worsened in the other. The frequency and magnitude of these potential manipulations were not specified (see Supplementary Materials for full instructions). In these trials, in addition to providing estimates, participants were asked to judge whether the feedback reflected their true performance or had been manipulated. Importantly, the feedback in the Agent condition was pre-programmed in the same way as in the corresponding categories without supposed manipulation (No Agent condition). This resulted in a 2×2 within-subject design with 20 trials per condition (Ability [High vs. Low] × Interference [Agent vs. No Agent], Fig. 1B). Trials from all conditions were intermixed in a fixed order, with no more than two consecutive trials of the same condition. This fixed sequence was used to ensure identical trial structure across participants, maintain a balanced distribution of conditions throughout the task, and provide reliable and comparable trial-by-trial signals for computational modeling, enabling robust estimation of learning rate parameters.

Instead of providing fixed feedback values, the feedback sequence was designed to elicit prediction errors of specific magnitudes and valences[20,21,33]. Planned prediction errors were drawn from a hand-designed sequence that followed a predefined distribution for each condition (high ability: −18 to 27, 70% positive, 30% negative; low ability: −27 to 18, 30% positive, 70% negative). The sequence was identical for all participants and followed a fixed order within each experimental condition. For each trial, the feedback value was calculated by adding the corresponding planned prediction error from the sequence to the participant's current ability belief, defined as the average of their last five expectation ratings within the respective condition. Before participants had provided their first performance expectation rating, the expectation value was set to 50%. This approach resulted in varying feedback sequences across participants while keeping actual prediction errors largely independent of individual performance expectations. It also ensured a relatively balanced distribution of actual negative prediction errors (Agent: $M = − 12.7$, 45.9% of trials; No Agent: $M = − 12.6$, 45.4%) and positive prediction errors (Agent: $M = 14.5$, 54.1%; No Agent: $M = 14.2$, 54.6%) across conditions.

Each trial started with a cue indicating the estimation category (e.g., height) and whether a manipulation of the feedback was possible (Fig. 1A). Participants were then asked to state how well they expected to perform in

Fig. 1 | **Overview of the learning of own performance (LOOP) task and task performance. A** In a within-subject design, participants were asked to estimate different properties in four distinct categories. A cue indicating the estimation category and whether a manipulation of the feedback was possible was followed by a performance expectation rating. The estimation question was subsequently presented for 10 s. Feedback was presented for 5 s after each estimation, indicating the participant's performance compared to an alleged reference group. In trials of the Agent condition, participants were asked whether they believed the feedback had been manipulated. **B** The four estimation categories were randomly assigned to four experimental conditions. Two categories were randomly paired with mostly positive feedback (High Ability) and two with mostly negative feedback (Low Ability). Participants were informed that in two categories, their feedback could be manipulated by the computer (Agent), such that feedback in one category might be improved and in the other worsened. In the other two categories, the feedback was allegedly not manipulated (No Agent). **C** Participants ($n = 64$) adjusted their performance expectations (solid lines) based on the feedback they received (blue: High Ability, red: Low Ability), with the adjustments varying depending on whether a manipulation was possible (dark color: No Agent, light color: Agent). Shaded areas represent the standard errors for the actual performance expectations for each trial. Our winning computational model captured the dynamics in participants' performance expectations (dashed lines).

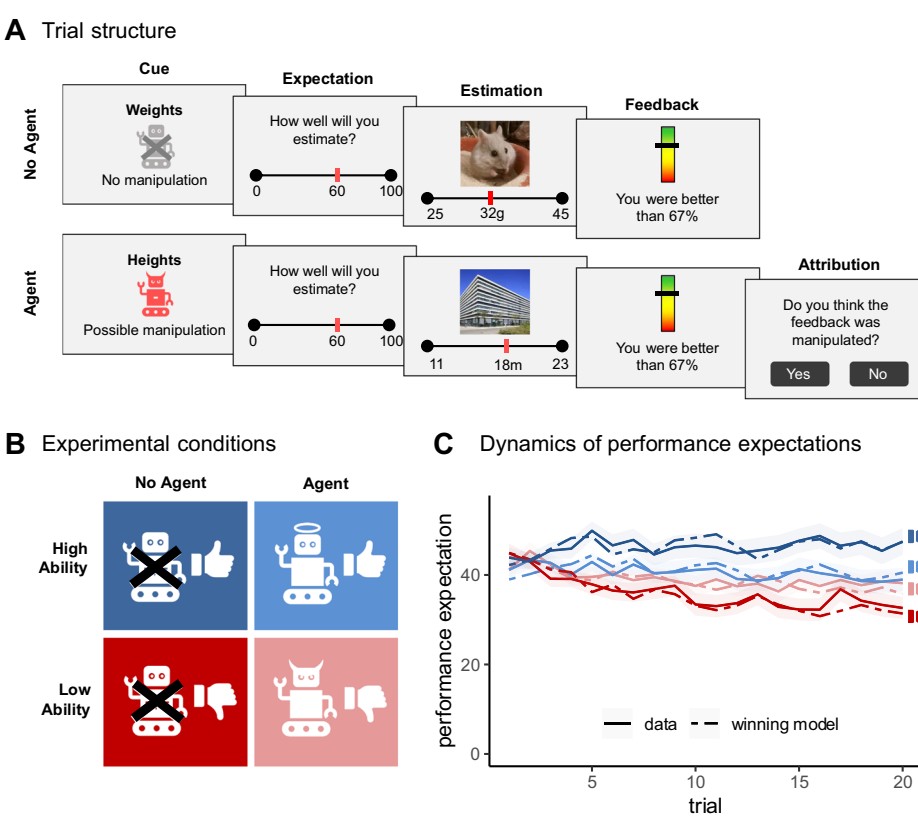

this trial in comparison to a reference group. Following each performance expectation rating, the estimation question was presented for 10 s. During the estimation period, continuous response scales below the pictures determined a range of plausible answers for each question. Performance feedback was provided for 5 s after each estimation, indicating the participant's accuracy as percentiles compared to an alleged reference group of 350 university students, who, according to the cover story, had been tested previously (e.g., 'You are better than 94% of the reference participants'). In trials of the Agent condition, participants were subsequently asked whether they believed the feedback they just received had been manipulated, thereby assessing whether they attributed the feedback to external or internal causes.

## Questionnaires and debriefing

Prior to the LOOP task, we collected general demographic information, assessed participants' self-beliefs regarding their estimation ability, and measured self-esteem. To assess self-esteem, participants completed the general self-concept scale from the Self-Description Questionnaire-III (SDQ-III)[36], which is based on the Rosenberg Self-Esteem Scale[37]. Depressive symptoms were assessed after the LOOP task using the Patient Health Questionnaire (PHQ-9). The PHQ-9 is the depression module of the PRIME-MD diagnostic instrument for common mental disorders and has been validated in the general German population[38–40]. After completing the LOOP task and questionnaires, participants took part in a brief follow-up survey assessing their impressions of the task. This survey included questions about the perceived plausibility of the feedback during the task, as well as participants' retrospective estimates of how frequently the computer manipulated the feedback (Supplementary Fig. S1). Although some participants expressed non-specific suspicions about the feedback, none explicitly indicated disbelief in the cover story, and no participants were excluded on this basis. Finally, participants were debriefed about the cover story and

reimbursed for their participation. During debriefing, they were informed that the feedback had been pre-programmed and did not reflect their actual performance, and the purpose of the manipulation was explained. The whole procedure, including instructions, the LOOP task, questionnaires, and debriefing, took approximately one hour.

## Statistical analysis

**Model-agnostic analysis**. A model-agnostic analysis of participants' performance expectations was conducted to illustrate basic task effects in our behavioral data. To this end, we fit a linear mixed-effects model predicting performance expectations from Trial (centered), Ability condition (High vs. Low), and Interference condition (Agent vs. No Agent), including all interactions. The model included random intercepts and random slopes for Trial across participants.

**Computational modeling**. Reinforcement learning equations based on Rescorla-Wagner were used to capture trial-by-trial changes in performance expectation ratings[41,42] (Fig. 2A). The most basic equation reads as follows (1):

$$EXP_{t+1} = EXP_t + \alpha \cdot PE_t \tag{1}$$

Here, "$EXP_{t+1}$" stands for the expectation rating of the next trial, "$EXP_t$" is the expectation rating of the current trial "$t$", and "$\alpha$" depicts the learning rate that functions as a weight on the prediction error "$PE$". The prediction error "$PE$" is the difference between the received feedback "$FB$" and the current performance expectation "$EXP_t$" (2):

$$PE_t = FB_t - EXP_t \tag{2}$$

**Fig. 2 | Computational modeling of self-belief formation. A** All computational models in our model space were built on Rescorla-Wagner delta-rule update equations, which modeled changes in performance expectations (EXP) by incorporating multiple learning rates (α, see Methods) and accounting for trial-by-trial prediction errors (PE) based on the provided feedback (FB). **B** In our models, we distinguished two factors that might influence learning rates and/or expectation updates in general: the Interference condition (Agent vs. No Agent) and trial-by-trial attributions (external vs. internal). As a baseline, we included a model that did not account for any of these factors (M1), but included separate learning rates for positive and negative prediction errors. While valence models (M2 and M4) incorporated separate valence-specific learning rates for each level of the factor of interest, weighting models (M3 and M5) included two valence-specific learning rates and an additional weight factor *s* that reduced the update depending on the attribution or interference condition (see Methods for details). The winning model is outlined in red.

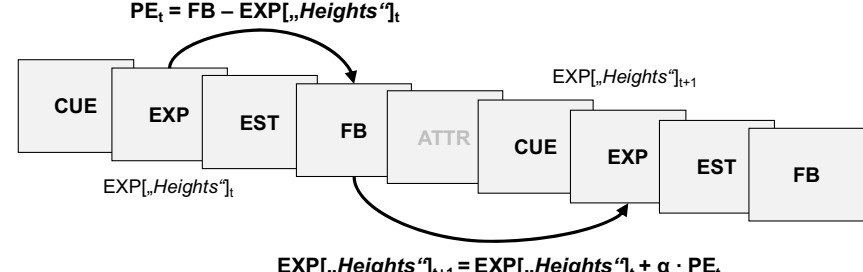

**A** Delta-rule update equation

$$PE_t = FB - EXP[„Heights"]_t$$

$$EXP[„Heights"]_{t+1} = EXP[„Heights"]_t + \alpha \cdot PE_t$$

**B** Model space and winning computational model

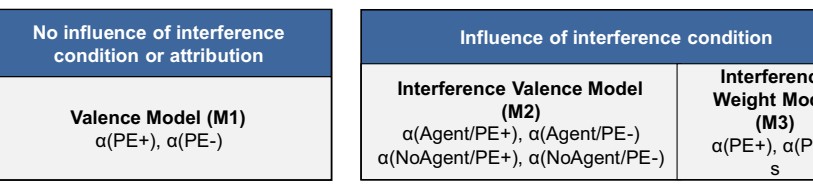

| No influence of interference condition or attribution |
| --- |
| **Valence Model (M1)**<br>α(PE+), α(PE-) |

| Influence of interference condition | |
| --- | --- |
| **Interference Valence Model (M2)**<br>α(Agent/PE+), α(Agent/PE-)<br>α(NoAgent/PE+), α(NoAgent/PE-) | **Interference Weight Model (M3)**<br>α(PE+), α(PE-)<br>s |

**Model equations of winning model:**

For *internal* attributions:
$$EXP_{t+1} = EXP_t + \alpha \times PE \times (1 - w \times ND)$$

For *external* attributions:
$$EXP_{t+1} = EXP_t + \alpha \times PE \times (1 - w \times ND) \times (1 - s)$$

| Influence of attribution | |
| --- | --- |
| **Attribution Valence Model (M4)**<br>α(Internal/PE+), α(Internal/PE-)<br>α(External/PE+), α(External/PE-) | **Attribution Weight Model (M5)**<br>α(PE+), α(PE-)<br>s |

Note that the trial index "*t*" and trial-by-trial changes of expectation ratings are always understood within the respective experimental condition, since belief formation is specific for the estimation categories that are linked to the combinations of the levels of Interference × Ability. The basic equation was expanded in two ways based on previous studies[20,21,33]: First, different learning rates were introduced to give varying weight to prediction errors based on their valence—differentiating between positive prediction errors ("better than expected") and negative prediction errors ("worse than expected"). Additionally, a parameter *w*—a feedback weight factor—was introduced to lessen the impact of feedback near the scale's limits. The latter was introduced based on the assumption that feedback that approaches the 0 or 100 percentiles may be less probable and therefore holds less informational value. This parameter *w* reduces the impact of prediction errors according to the feedback percentile. For this, it is multiplied by the relative probability density of the normal distribution "*ND*" of the possible feedback values, culminating in the extended Eq. (3) that serves as the base model for all models included in our model space:

$$EXP_{t+1} = EXP_t + \alpha_{valence} \cdot PE_t \cdot (1 - w \cdot ND) \qquad (3)$$

The final model space (Fig. 2B) contained five models. The first model (M1) assumed no influence of the Interference condition or attribution and only differentiated two learning rates for positive vs. negative prediction errors, as shown in the equation above. All other models considered the Interference condition, but differed in their approach. A first set of two models did so by including the Inference condition, but without considering individual attributions per trial. The Interference Valence Model (M2) proposed four distinct learning rates based on prediction error valence and the level of Interference. The Interference Weighting Model (M3) assumed only two learning rates, separated by valence, but introduced an additional parameter *s*. This parameter also decreased the update, but only in trials of the Agent condition. The second and final set of two models assumed an

influence of trial-by-trial attributions on updating, whether subjects believed an agent had manipulated the feedback. The Attribution Valence Model (M4) considered four learning rates based on prediction error valence and whether or not participants indicated that an agent had manipulated the feedback in the respective trial. The Attribution Weighting Model (M5), proposed only two learning rates separated by prediction error valence and again considered a parameter *s*. In this model, the parameter *s* exclusively decreased learning rates for external attributions, so the model's equations differed for trials with internal (4) vs. external attributions (5):

for internal attributions:

$$EXP_{t+1} = EXP_t + \alpha_{valence} \cdot PE_t \cdot (1 - w \cdot ND) \qquad (4)$$

for external attributions:

$$EXP_{t+1} = EXP_t + \alpha_{valence} \cdot PE_t \cdot (1 - w \cdot ND) \cdot (1 - s) \qquad (5)$$

For trials of the no agent condition, where no attribution ratings were available, we assumed internal attributions throughout.

Based on previous studies that used the LOOP task, for this study, we deliberately did not consider simpler models that assume a singular learning rate across all conditions or different learning rates for each level of the factor Ability, since these model variants were regularly outperformed by models that differentiate learning rates for positive vs. negative prediction errors[18,20,21,33,35].

**Model fitting.** For model fitting, the *rstan* package[43] for the software *R* was utilized. For all subjects, models were fitted individually with Markov Chain Monte Carlo (MCMC) sampling. Four MCMC chains were used, and 3400 samples (thereof 1000 burn-in samples) were drawn, thinned by a factor of three. $\hat{R}$ values for each parameter were inspected to check the convergence of the MCMC chains. Furthermore, the effective sample

sizes ($n_{eff}$) were verified to generally exceed 1500. This was done to check whether there were sufficient numbers of independent draws from the posterior distributions of the model's parameters. After the successful model estimation, mean values per parameter and subject were calculated to summarize the posterior distributions.

**Model selection**. To determine which model best assessed the subjects' updates of performance expectations, we estimated the pointwise out-of-sample accuracy for all fitted models within the model space, separately for each subject. For this, leave-one-out cross-validation (*LOO*; in this case, one trial per subject was left out) was utilized, and Pareto-smoothed importance sampling (*PSIS*) was applied by using the log-likelihood that was calculated from the posterior simulations of the estimated parameters[44]. Sum *PSIS-LOO* scores for each model were used for model comparison (Supplementary Table S1). In addition, $\hat{k}$ values were inspected - the estimated shape parameters of the generalized Pareto distribution. These values indicate the reliability of the *PSIS-LOO* estimates, and very few trials resulted in values that indicate unreliable scores ($\hat{k} > 0.7$, Supplementary Table S1). Further, Bayesian model selection (BMS) using *PSIS-LOO* values was conducted to identify the model that best described participant behavior at the sample level, considering the heterogeneity among participants. This provided the protected exceedance probability (*pxp*) for each model considered. The *pxp* is a metric that indicates the likelihood of a given model explaining the data in comparison to all other models in the model space. Additionally, it included the Bayesian omnibus risk (*BOR*), which serves as the posterior probability that the frequencies of all the considered models are equal (see Supplementary Fig. S2 for BMS results).

**Posterior predictive checks**. To assess if the winning model captured the core effects in participant behavior, we repeated the model-agnostic analysis conducted on actual performance expectations with the data predicted by the winning model as a posterior predictive check. Specifically, we fit a linear mixed-effects model on predicted performance expectations from Trial, Ability condition, and Interference condition, including all interactions. Again, the model included random intercepts and random slopes for Trial across participants.

**Analysis of model parameters**. Model parameters, that is, learning rates and attribution weight factors, of the winning model were analyzed on the group level using *R* Version 4.1.3. We compared learning rates for positive and negative prediction errors by calculating a signed Wilcoxon rank test. Attribution weight factors were compared against 0 using a one-sample *t*-test to demonstrate that attributions of feedback significantly influenced updates in performance expectations. Attribution weight factors of 0 indicate that causal attributions of the received feedback (internal vs. external) do not affect self-belief formation.

To investigate the associations of depressive symptoms and self-esteem with a valence bias in learning, we calculated a bias score for each subject. This was done by dividing the difference between learning rates for positive and negative prediction errors by their sum[18,20,33]:

$$learning\ bias = (\alpha_{PE+} - \alpha_{PE-}) \div (\alpha_{PE+} + \alpha_{PE-}) \qquad (6)$$

Spearman correlations were then calculated between individual bias scores and PHQ-9 and SDQ-III sum scores, respectively. To account for multiple hypothesis testing, *p*-values were adjusted using the Benjamini-Hochberg false discovery rate (FDR) procedure.

**Analysis of feedback attributions**. To quantify a potential attributional bias in the Agent condition—the tendency to attribute better-than-expected feedback to one's own ability while attributing worse-than-expected feedback to manipulation by the computer agent—we tested whether attribution patterns varied as a function of prediction error valence. Because feedback values closer to the ends of the feedback scale

may be perceived as less plausible and could therefore increase external attributions independent of prediction error valence, feedback extremity was included as a control predictor.

To this end, we fit two linear mixed-effects models predicting the proportion of external attributions. Both models included feedback extremity decomposed into within-subject and between-subject components to distinguish trial-by-trial fluctuations in feedback extremity from individual differences in average feedback extremity. Prediction error valence and participants' depressive symptoms or self-esteem (mean-centered) were included as predictors, as well as their interaction. Due to the high correlation between depression and self-esteem, these variables were entered in separate models to avoid multicollinearity. Both models included random intercepts for participants and random slopes for prediction error valence by participant. Random slopes for feedback extremity were initially considered but were not included in the final model because their inclusion led to model non-convergence. For each fixed effect of interest, *p*-values obtained from the two linear mixed-effects models were jointly adjusted for multiple comparisons using the FDR procedure, such that corresponding effects (e.g., main effect of prediction error valence, interaction with depression/self-esteem) were corrected together.

**Preregistration**. This study's design and general research questions were preregistered on AsPredicted before data collection (aspredicted.org/n75w-npng.pdf, preregistered on 01/25/2024). Although the general study design remained unchanged, it was necessary to make some adjustments in the testing of hypotheses, as the use of a more complex computational model was found to be more effective. The preregistered methods proposed a computational model that incorporates four separate learning rates that allow the analysis of participants' updating behavior in each experimental condition. This model was included in the model space as M2 (see Fig. 2B). However, because it was outperformed by an alternative model with a different set of parameters, the originally planned analyses examining four separate learning rates could not be conducted. Consequently, our analyses instead included the set of parameters of the winning model, that is, two learning rates for positive and negative prediction errors across all experimental conditions, as well as an additional attribution weight factor. Following this logic, we also examined the proportion of external attributions by prediction error valence to assess potential attributional biases, and not, as preregistered, by experimental condition. Further, self-esteem was included as an additional independent variable to complement the score on depressive symptom severity. Instead of including self-esteem and depressive symptom severity as covariates in the analyses of learning rates, correlations with the scores for learning biases were calculated. Because of these deviations from the original analysis plan, we conducted a post-hoc sensitivity analysis to determine the minimum effect size that could be reliably detected within our sample (Supplementary Note 2).

## Results
### Model-agnostic behavioral analysis
A linear mixed-effects model predicting performance expectations from Trial, Interference condition, and Ability condition revealed a small but significant decrease in performance expectations across trials ($\beta = -0.2$, 95% CI [$-0.35$, $-0.04$], t(63) = $-2.47$, $p = 0.016$). Across trials and Ability conditions, expectations were lower in the Agent condition compared to the No-Agent condition ($\beta = -1.45$ [$-2.08$, $-0.82$], t(4986) = $-4.53$, $p < 0.001$). Expectations were substantially higher for the High Ability compared to the Low Ability condition ($\beta = 5.8$ [$5.17$, $6.43$], t(4986) = 18.11, $p < 0.001$).

The interaction between Trial and Interference was not significant ($\beta = -0.02$ [$-0.13$, $0.09$], t(4986) = $-0.31$, $p = 0.753$), suggesting that expectation change was comparable between the Agent and No-Agent conditions irrespective of feedback valence. In contrast, the interaction between Trial and Ability was significant ($\beta = 0.31$ [$0.2$, $0.42$], t(4986) = 5.55, $p < 0.001$), indicating that the change in performance expectations across

trials differed between the High and Low Ability conditions. There was also a significant interaction between Interference and Ability ($\beta = -8.83$ [$-10.09$, $-7.58$], t(4986) = $-13.79$, $p < 0.001$).

Importantly, these effects were further qualified by a significant three-way interaction between Trial, Interference, and Ability ($\beta = -0.52$ [$-0.74$, $-0.3$], t(4986) = $-4.67$, $p < 0.001$). This interaction indicates that belief updating over time differed depending on both the valence of feedback and the attributional context. To further decompose the three-way interaction, we examined expectation change across trials separately for each condition (Fig. 1C). In the No Agent condition, participants showed pronounced updating, with expectations increasing over time in the High Ability condition ($\beta = 0.10$ [$-0.09$, 0.28], $p = 0.301$) and decreasing in the Low Ability condition ($\beta = -0.47$ [$-0.65$, $-0.29$], $p < 0.001$). In contrast, in the Agent condition, expectations decreased in both Ability conditions, though effects were weaker and only marginal in the High Ability condition (Low Ability: $\beta = -0.23$, SE = 0.09, 95% CI [$-0.41$, $-0.05$], $p = 0.014$; High Ability: $\beta = -0.18$, SE = 0.09, 95% CI [$-0.36$, 0.00], $p = 0.052$). Follow-up comparisons showed a significant difference between High and Low Ability in the No-Agent condition ($\Delta\beta = -0.57$, 95% CI [$-0.72$, $-0.41$], $p < 0.001$), but not in the Agent condition ($\Delta\beta = -0.05$, 95% CI [$-0.20$, 0.10], $p = 0.536$), indicating that feedback-dependent differences in updating were attenuated when outcomes could be attributed to an external agent. This pattern suggests that expectation updating is shaped by valence-dependent prediction errors and modulated by attributional processes, providing the basis for the computational modeling approach.

### Selection of computational models of self-belief formation
To account for the observed interaction between trial, ability, and interference, we implemented a computational model of belief updating that captures how participants adjust their performance expectations over time as a function of feedback valence and attributional context. The model that best described the observed behavior contained separate learning rates ($\alpha$) for updates following positive vs. negative prediction errors (Fig. 2B). This allowed a prediction-error valence-based description of belief formation. Further, the winning model included an attribution weight factor, $s$, that scales how strongly participants learn from externally attributed feedback, relative to the rate at which they learn from internally attributed feedback. An $s$ factor closer to 0 indicates that on trials where subjects attribute feedback externally, they learn at a similar rate as on trials where feedback is attributed internally. Inversely, an $s$ factor of (close to) 1 indicates that subjects do not learn from externally attributed feedback at all, thus maintaining their current self-belief (see Supplementary Fig. S3 for an illustration of this effect). The model estimation resulted in largely uncorrelated estimates of learning rates and the parameters $w$ and $s$ (see Supplementary Table S3).

Posterior predictive checks validated that the winning model accurately captured the key effects from our model-agnostic analysis, showing that the feedback and interference conditions jointly influenced dynamics in predicted performance expectations (see Supplementary Table S2 and Fig. 1C).

### Negativity bias in self-belief formation and reduced updating after externally attributed feedback
On average, participants showed higher learning rates for negative ($Mdn = .24$) than for positive prediction errors ($Mdn = 0.11$) (Wilcoxon signed rank test, z(63) = -3.71, $p < 0.001$, r = 0.46, Hodges-Lehmann estimate = 0.14, 95% CI [0.07,0.22]), suggesting a general negativity bias during self-belief formation over all experimental conditions (Fig. 3A). Attribution weight factors were significantly greater than 0 with a mean value of $M = 0.51$ (SD = 0.17, t(63) = 24.3, $p < 0.001$, d = 3.04 [2.31, 3.77], Fig. 3B). Thus, on average, updates in performance expectations were reduced by half after externally attributed feedback compared to internally attributed feedback. In other words, participants integrated feedback into their self-beliefs to a lesser extent when they believed the computer had manipulated it.

### Depression and self-esteem are associated with biased learning
To assess the relationship of depressive symptoms and self-esteem with a valence bias in learning, we calculated individual bias scores for each participant. Correlation analyses showed a negative association between bias scores and depressive symptoms (Spearman $\rho = -0.26$, 95% CI [$-0.47$, $-0.01$], $p_{FDR} = 0.04$) as well as a positive association between bias scores and self-esteem ($\rho = 0.44$ [0.21, 0.62], $p_{FDR} < 0.001$, Fig. 3D). This suggests that individuals with elevated depressive symptoms and lower self-esteem showed a stronger negativity bias in learning. Depression and self-esteem were strongly negatively correlated ($\rho = -0.62$ [$-0.75$, $-0.45$], $p_{FDR} < .001$).

### Self-serving attributional bias and its relationship with depression and self-esteem
We examined how the proportion of external attributions varied with prediction error valence and participants' depressive symptoms and self-esteem, controlling for feedback extremity. To this end, we fit two separate linear mixed-effects models that were identical except that one included depression and the other included self-esteem as a predictor. The results of both models are shown in Fig. 4.

Trial-by-trial feedback extremity consistently predicted external attributions across both models, with more extreme feedback associated with more external attributions (depression model: $\beta = 0.0063$, 95% CI [0.0045, 0.0081], t(937) = 6.95, $p < 0.001$; self-esteem model: $\beta = 0.0064$ [0.0045, 0.0081], t(974) = 6.98, $p < 0.001$, see Fig. 4A, B for standardized beta coefficients). Between-subject differences in average feedback extremity were not significant in either model (depression model: $\beta = 0.0018$ [-0.0045, 0.0081], t(70) = 0.55, $p = 0.585$; self-esteem model: $\beta = 0.0000$ [-0.0068, 0.0068], t(66) = 0.00, $p = 0.997$).

Prediction error valence significantly predicted external attributions in both models, with worse-than-expected feedback associated with more external attributions (depression model: $\beta = -0.0547$ [$-0.1024$, $-0.0066$], t(70) = $-2.23$, $p_{FDR} = 0.029$; self-esteem model: $\beta = -0.0547$ [$-0.101$, $-0.0082$], t(71) = $-2.31$, $p_{FDR} = 0.029$). That is, participants were more likely to attribute feedback to a manipulation by the computer agent when the feedback was worse than expected than when it was better than expected (see Fig. 3C for raw data). This corresponds to a self-serving attributional bias.

Depressive symptoms were negatively associated with external attributions ($\beta = -0.0075$ [$-0.0129$, $-0.002$], t(63) = $-2.67$, $p_{FDR} = 0.019$), but the interaction between prediction error valence and depressive symptoms was not significant ($\beta = 0.0045$ [$-0.0034$, 0.0124], t(63) = 1.11, $p_{FDR} = 0.269$). In contrast, self-esteem alone was not significantly associated with external attributions ($\beta = 0.0011$ [$-0.0039$, 0.0062], t(65) = 0.44, $p_{FDR} = 0.665$), but the interaction between prediction error valence and self-esteem was significant ($\beta = -0.0078$ [$-0.0142$, $-0.0013$], t(68) = $-2.34$, $p_{FDR} = 0.045$). This indicates that the effect of prediction error valence on external attributions was stronger among participants with higher self-esteem; in other words, participants with higher self-esteem showed a stronger self-serving bias (Fig. 4B).

### Discussion
The present study examined how momentary causal attributions shape the formation of self-related ability beliefs. Specifically, we assessed whether attributions of feedback in a validated learning task influence the process of belief formation, and whether individual differences in self-esteem and depressive symptoms are associated with systematic biases in attributions and learning. Moving beyond trait-like conceptualizations of attributional style, this approach captures how causal interpretations dynamically unfold and shape learning about one's own abilities in real time. Our findings provide three main insights. First, using computational modeling, we show that momentary causal attributions systematically modulate self-related learning. When feedback was attributed to an external agent, belief updating was attenuated, indicating that participants adjusted their self-beliefs to a lesser degree when they thought that the feedback had been manipulated. Second, we observed a negativity bias in self-related learning, showing that

Fig. 3 | **Parameters of winning model, percentage of external attributions made during the task, and associations of bias in learning with depression and self-esteem (n = 64 participants). A** Learning rates by prediction error (PE) valence (jittered raw data, boxplots and probability distributions). Learning rates indicate stronger updates in performance expectations after negative ("neg", red) vs. positive ("pos", blue) prediction errors. **B** Attribution weight factors (jittered raw data, boxplots and probability distributions). Attribution weight factors were significantly greater than zero, leading to diminished updates after external vs. internal attributions. **C** Percentage of external attributions by prediction error (PE) valence (jittered raw data, boxplots and probability distributions). The percentage of external attributions was higher after negative ("neg", red) vs. positive ("pos", blue) prediction errors. **D** Higher depression levels and lower self-esteem were associated with a stronger negativity bias in learning. Shaded areas indicate the 95% confidence interval around the estimated regression line (red).

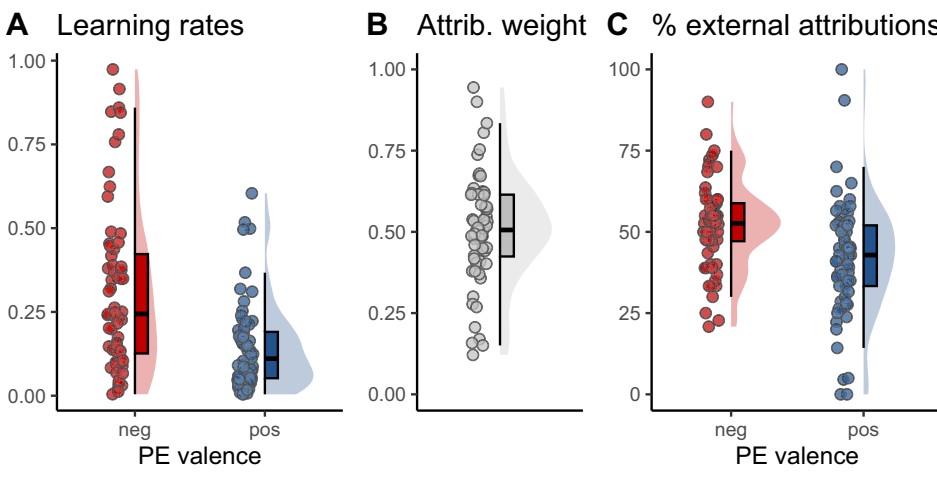

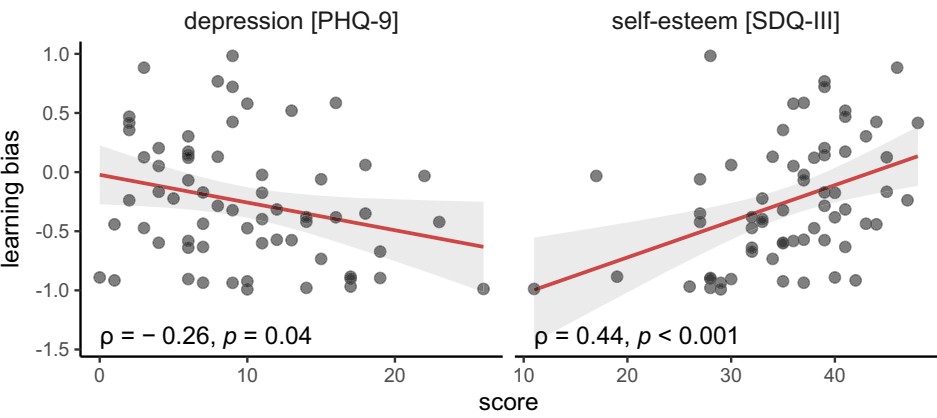

participants adjusted their ability beliefs to a stronger extent after worse-than-expected feedback, compared to better-than-expected feedback. This negativity bias was more pronounced in individuals with higher levels of depression and lower self-esteem. Third, our results suggested a self-serving attributional pattern, such that worse-than-expected feedback was more likely attributed to external causes, whereas better-than-expected outcomes were attributed to one's own ability. Less self-serving attributional tendencies were specifically associated with lower self-esteem, rather than depressive symptom severity.

These findings extend prior work on causal attributions and associated biases by manipulating and assessing trial-by-trial causal attributions during self-related ability belief formation. Previous studies have demonstrated that people tend to attribute positive outcomes to themselves and negative outcomes to external causes. This could be shown when learning about reward probabilities in a forced-choice task[28,29], when attributing losses in a gambling task[45], and even in unambiguous contexts with clear clues for causality[46]. However, these studies mainly focused on attributions of events that reflected beliefs about the world, such as causes of gains or losses in reward learning. By examining causal attributions in learning about oneself, we show that causal attributions are not only related to past outcomes but also dynamically modulate the formation of self-related ability beliefs in real-time. Importantly, our results highlight two central biases: A negativity bias in learning, reflected by an overemphasis on negative prediction errors compared to positive prediction errors, and a self-serving attributional bias, in which unfavorable results are more likely attributed to external causes. Notably, while both biases are likely tied to self-referential processing[18,21,33,47,48], the computational mechanism captured by our model – reduced updating from outcomes perceived as externally caused – may reflect a more general learning principle that is not limited to self-related contexts.

Although both the negativity bias[18,20,21,33] and the self-serving attributional bias[28,29,49] have been documented independently in healthy populations, our results indicate that they can both affect self-related ability beliefs within the same context, while remaining analytically separable. In our task, the possibility of external interference specifically introduced uncertainty about the validity of the feedback in the Agent condition. Under these circumstances, participants may have used external attributions as a cognitive safeguard, treating outcomes as potentially not fully reflective of their own performance. As implied by the structure of our winning computational model, this attributional tendency selectively attenuated the integration of feedback, resulting in relatively stable performance expectations in the Agent condition. This effect was more pronounced for worse-than-expected outcomes, which were more likely to be attributed to external causes under the self-serving attributional bias. In contrast, across all conditions, we observed a general negativity bias in learning, whereby worse-than-expected feedback exerted a stronger influence on belief updating than better-than-expected feedback. Thus, while the negativity bias shaped belief updating overall, external attributions in the Agent condition reduced the extent to which such negative feedback was incorporated. Taken together, in contexts with ambiguous sources of self-related feedback, the attributional bias may support the maintenance of desirable self-beliefs by strengthening self-efficacy and self-coherence[26,27,50]. In parallel, the negativity bias in learning may serve to guard against overly optimistic or potentially erroneous belief updates.

Our results suggest that these two biases are associated with self-esteem. Participants with higher self-esteem expressed a lesser negativity bias in learning, as was the case in previous studies[18,20]. Higher self-esteem appears to protect against an overemphasis on worse-than-expected feedback. At the same time, individuals with higher self-esteem demonstrated a stronger self-serving bias in attribution, as they were more inclined to

**Fig. 4 | Proportion of external attributions as a function of prediction error valence and depression/self-esteem (n = 64 participants).**
**A** Depression and external attributions. Left: standardized beta coefficients of a linear mixed model on the proportion of external attributions, including depression (PHQ-9). Results show a significant main effect of prediction error (PE) valence indicating a self-serving attributional bias, as well as a significant main effect of depression. Right: illustration of the main effects of depression and prediction error (PE) valence. Shaded areas indicate 95% confidence intervals. **B** Self-esteem and external attributions. Left: standardized beta coefficients of a linear mixed model on the proportion of external attributions including self-esteem (SDQ-III). Results show a significant interaction of self-esteem and PE valence. Right: illustration of interaction effect of self-esteem and PE valence. Shaded areas indicate 95% confidence intervals.

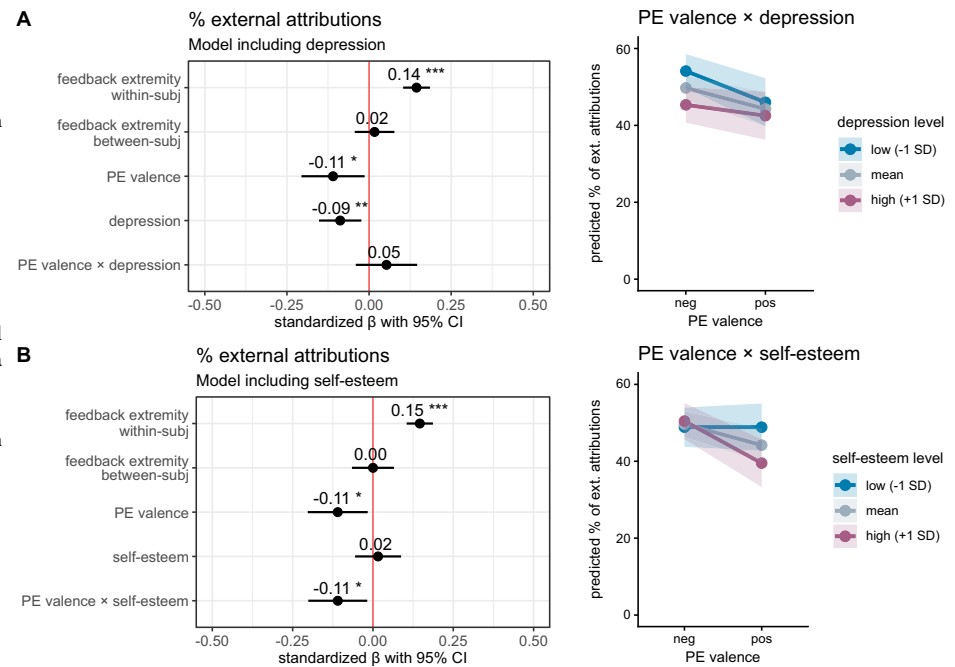

attribute negative outcomes to external factors. For these individuals, this tendency resulted in diminished updates to their ability beliefs after worse-than-expected feedback, which could have otherwise shifted their beliefs in unfavorable directions. From this perspective, again, self-serving attributions appear ambivalent: they constrain learning but serve an adaptive function by stabilizing desirable self-belief systems[26,27,50,51]. These findings align with social-cognitive theories[26,27,52–54], which propose reciprocal reinforcement between attributional styles and self-beliefs. Such recursive dynamics may not only explain the persistence of self-serving views but also the persistence of pessimistic self-views: external attributions of positive outcomes impede corrective learning, and additionally, negative self-beliefs increase the likelihood of maladaptive, pessimistic attributions that, in the long run, may impair self-esteem[2,6].

Notably, self-serving motives—those to preserve self-efficacy, self-coherence, and positive affect—are also discussed in the context of the optimism bias in belief updating that was often shown in other domains of self-referential learning: the tendency of updating expectations about future events in an optimistically biased way[24,55]. This supports the notion that attributional processes play an important role in the formation and updating of (self-)beliefs, especially in their role to protect desirable self-beliefs.

Contrary to our hypothesis, attributional bias was not significantly associated with depressive symptoms. However, several explanations are possible. First, the predominantly subclinical symptom levels in our sample may have had restricted variance and limited power to detect such associations.

More importantly, depressive symptoms and self-esteem may reflect partly distinct psychological processes with different relevance for attributional style. Depression, as assessed using the PHQ-9, captures a broad and heterogeneous constellation of affective, cognitive, and somatic symptoms[39], many of which may not be directly linked to causal attributions of success and failure. By contrast, self-esteem reflects global self-evaluative beliefs[56] and may therefore be conceptually more proximal to the ability-related self-beliefs examined in the present study[7,8,51,57]. From this perspective, the reduced self-serving bias commonly described in depression[2,5,6] may be more strongly related to low self-esteem than to depressive symptom severity per se[9] and may therefore also extend to other forms of psychopathology characterized by diminished self-esteem, such as anxiety disorders[58,59]. Future studies comparing depressive and non-depressive

clinical populations may help disentangle the relative contributions of depressive symptoms and self-esteem to attributional biases.

## Limitations
While the present findings provide initial insights into the role of causal attributions in the process of forming new ability self-beliefs, several considerations are important for their interpretation. First, the sample size may limit statistical power and the robustness of the observed effects. A sensitivity analysis indicated that the present design was primarily powered to detect medium-sized associations, suggesting that smaller effects may not have been reliably detected. Accordingly, the observed associations between self-esteem and attributional biases should be interpreted with some caution and warrant further investigation in larger samples to establish their stability and generalizability. Second, given that the study was conducted in a student sample, depressive symptom levels were likely to fall predominantly below clinical thresholds, and participants' diagnostic status was not assessed. As a result, effects involving depressive symptoms may be underestimated due to limited between-subject variability in PHQ-9 scores. Stronger or qualitatively different effects might emerge in samples that include individuals with a clinical diagnosis of depression. Third, given that depressive symptoms and self-esteem were not experimentally manipulated, the directionality and causality of their associations with attributional patterns remain unclear. Theories assume recursive dynamics between aspects of the self-concept and attributional patterns[26,27,52–54]. While our current analysis demonstrates self-serving attributional patterns and shows how these may influence belief updating over time, it does not allow conclusions about the causal interplay between the resulting beliefs and self-esteem or depressive symptoms, nor about whether such beliefs subsequently shape future attributions. Again, the relatively small sample size may additionally constrain the ability to test more complex reciprocal or directional effects.

Despite these issues, the present findings contribute preliminary evidence that is consistent with contemporary psychological theories[26,27] and previous findings in related fields[28,29,31], and may help to set a direction for future studies examining attributional patterns and effects on self-belief formation.

## Conclusion and outlook
Our findings suggest that self-related ability beliefs in a performance context can be shaped by two distinct, yet concurrently occurring biases. The first is a

negativity bias in learning, meaning a greater emphasis on negative prediction errors compared to positive ones. The second is a self-serving attributional bias, characterized by external causal attributions of worse-than-expected feedback, which is linked to self-esteem. Individuals with lower self-esteem, who disproportionately attribute negative outcomes to their own abilities, may therefore be especially prone to developing enduring negative self-related ability beliefs.

Future research should extend these findings to clinical samples with diagnosed major depression to further establish the relevance of attributional styles in self-related belief formation for psychopathology and therapeutic interventions. More specifically, interventions that target attributional styles, for instance, by addressing pessimistic and dysfunctional attributions, may enhance corrective learning about self-related concepts and abilities and thereby reduce vulnerability to depression.

## Data availability
The behavioral data and computational modeling results are openly available on OSF. All variables that could be used to identify individual participants (e.g., sociodemographic data) were removed from this data set.

## Code availability
The analysis code is openly available on OSF.

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

## Acknowledgements

We want to thank Alicia Kroell for her invaluable assistance with data collection.

## Author contributions

A.V.M., A.S., D.S.S., N.C., S.K., and T.K. designed the research. A.S. and A.V.M. realized the experimental task and supervised data acquisition. A.V.M., A.S., and D.S.S. analyzed the data. A.V.M. and A.S. prepared the manuscript. A.V.M., A.S., D.S.S., N.C., F.M.P., S.K., T.K., and L.M.P. discussed the data analyses and interpretation of the results and reviewed and edited the manuscript.

## Funding

The research was funded by the German Research Foundation (Project-based funding for TK: KU 3955/3-1 and SK: KR 3803/11-1; KR 3803/14-1; Temporary Position for Principal Investigator LMP: MU 4373/1-1; MU 4373/1-3) and the Medical Department of the University of Lübeck (Clinician Scientist funding for AVM: CS08-2023). The funders had no role in study design, data collection and analysis, decision to publish, or preparation of the manuscript. Open Access funding enabled and organized by Projekt DEAL.

## Competing interests

The authors declare no competing interests.
