## [Transparent Peer Review file · Communications Psychology]

Causal attributions shape the formation of novel ability self-beliefs

Corresponding Author: Dr Annalina Mayer

Version 0:

Decision Letter:

Dear Dr Mayer,

Thank you for your patience during the peer-review process. Your manuscript titled "Self-esteem modulates beneficial causal attributions in the formation of novel self-beliefs" has now been seen by 2 reviewers, and I include their comments at the end of this message. They find your work of interest but raised some important points. We are interested in the possibility of publishing your study in *Communications Psychology*, but would like to consider your responses to these concerns and assess a revised manuscript before we make a final decision on publication.

We therefore invite you to revise and resubmit your manuscript, along with a point-by-point response to the reviewers. Please highlight all changes in the manuscript text file.

Editorially, we identify two methodological issues that require careful and substantive attention.

First, all deviations from the preregistered plan must be clearly documented and justified (see Willroth & Atherton, 2024, doi: 10.1177/25152459231213802). Notably, the primary hypothesis tested in the manuscript differs from that specified in the preregistration and therefore requires a clear and compelling justification. In addition, replacing the preregistered GLM with questionnaire scores as covariates with a repeated-measures ANOVA constitutes a major analytic deviation that is currently insufficiently justified.

Second, both reviewers raised concerns regarding inflated false-positive risk due to multiple comparisons. To assess the evidential strength of the reported effects, I conducted a preliminary Bayes factor calculation based on the reported sample size and correlation coefficients using JASP's summary-statistics module (Aczel et al., 2018, doi: 10.1177/2515245918773742). These calculations indicate weak evidential support for the reported correlations. Furthermore, the preregistration assumed a medium effect size for sample-size planning without adequate justification; a sensitivity analysis is therefore required. Given the weak evidence currently available, additional data collection may be necessary to provide sufficiently robust support for the claimed correlations.

As you revise the manuscript in response to these issues, please also implement all requests in the attached Mandatory Revision Requests document. All requirements listed in this document need to be fully met, or the work will be returned to you for further revisions without peer review. This workflow is in place to increase the likelihood that the paper will be accepted for publication. It reduces the number of rounds of revision (and review) and ensures that the reviewers vet a version of the article that is compliant with journal policies. If you have any questions regarding the required revisions, please contact the journal prior to resubmission to avoid a negative outcome.

Please submit the following items:

- Revised manuscript

- Point-by-point response to the referees' comments
- Mandatory Revision Requests Table (attached).
- Cover letter (as a separate document)

via this link: Link Redacted .

** This url links to your confidential home page and associated information about manuscripts you may have submitted or are reviewing for us. If you wish to forward this email to co-authors, please delete the link to your homepage first **

Best regards,

Hu Chuan-Peng

Hu Chuan-Peng, PhD
Editorial Board Member
Communications Psychology
orcid.org/0000-0002-7503-5131

REVIEWER EXPERTISE:

Self-concept; self-belief, depression, computational modelling

REVIEWER REPORTS:

Reviewer #1 (Remarks to the Author):

This study examines how causal attributions of performance feedback influence self-related learning and the formation of self-beliefs, and how these processes relate to depression and self-esteem. Using a LOOP task combined with a computational model incorporating prediction error valence, causal attributions, and weight factors, the results show that belief updating is influenced when feedback is attributed to external causes. In addition, higher levels of depressive symptoms and lower self-esteem are associated with a stronger negativity bias in learning, while lower self-esteem is linked to reduced attributional bias. Overall, this is a well-designed and interesting study, and the comments below are intended to help improve clarity, transparency, and alignment between claims and evidence.

Introduction

The Introduction addresses a relevant and timely topic, namely the role of causal attributions in self-related learning and their relation to depression and self-esteem. The general thematic scope is appropriate for Communication Psychology, and the manuscript engages with both classical attribution theory and more recent computational perspectives. However, in its current form, the Introduction would benefit from clearer logical organization and a more coherent progression toward the study aims.

1. The articulation of the theoretical gap requires clarification. The manuscript notes that prior work has relied largely on self-report measures and that less is known about real-time causal attributions during self-related learning. This is a potentially strong and important gap; however, it is not formulated sharply enough. In particular, the contrast between what is known (e.g., attributional styles assessed retrospectively or at a trait level) and what remains unknown (e.g., trial-by-trial causal attributions interacting with belief updating) should be made more explicit and systematic.

2. The flow of the Introduction is weakened by redundancy and shifts in focus. Statements regarding negative biases in self-related learning in depression and low self-esteem appear both before and after the research questions are introduced, which disrupts the logical progression. Relatedly, some literature review elements appear after the explicit statement of research aims, whereas they would more naturally precede and motivate these aims.

3. The research questions themselves are relevant and clearly stated, but they are introduced somewhat abruptly. The Introduction would benefit from a more gradual build-up toward these questions, ensuring that the reader clearly understands why these questions logically follow from the preceding theoretical discussion.

Methodological Clarifications and Requests

4. Clarification of the cover story and agent manipulation

The following points relate to the transparency and validity of the cover story and feedback manipulation.

The manuscript refers to a “cover story” involving possible manipulation of feedback by a computer agent, but the details of this cover story are not sufficiently described. It remains unclear how the possibility of computer interference was explained to participants, why such manipulation was introduced, and how it was justified within the task narrative. Providing a clearer description of the cover story is important for evaluating participants’ understanding of the task and the validity of the attribution manipulation.

5. Belief in the cover story and participant exclusion

Previous work using similar paradigms (e.g., studies by Laura Müller-Pinzler et al.) has reported that some participants did not believe the cover story and were therefore excluded from analyses. It would be important to clarify whether a similar issue occurred in the present study. Specifically, were participants’ beliefs in the cover story assessed, and were any participants excluded due to explicit disbelief or suspicion regarding the agent manipulation?

6. Precision of performance expectation ratings

Based on the information of the figure, participants were not shown their exact numerical performance expectation ratings (e.g., percentile values) during the task. If these values were not explicitly displayed, participants’ expectations may have been relatively imprecise compared to the explicit and numerical feedback provided. Please clarify whether potential imprecision in expectation estimates could have influenced how participants perceived and interpreted discrepancies between expectations and feedback.

7. Specification of feedback magnitudes and distribution across conditions

While it is understandable that the Ability condition was manipulated via predominantly positive versus predominantly negative feedback, additional detail is needed regarding the exact structure of the feedback. Specifically, how large were the deviations between feedback and expectations (i.e., the magnitude of prediction errors), and did these deviations fall within a predefined range? Furthermore, please clarify how many positive versus negative feedback trials were presented within each condition (high vs. low ability), and whether this distribution was identical across participants.

8. Assessment of belief in agent interference during debriefing

Relatedly, the debriefing section does not report to what extent participants believed that their feedback had actually been manipulated by the agent. If participants’ beliefs in agent interference were measured explicitly (e.g., via post-task questions or ratings), reporting this information would substantially strengthen the interpretation of the attribution manipulation. If such measures were not collected, this limitation should be acknowledged.

9. Use of repeated-measures ANOVA versus linear mixed-effects models

The manuscript reports a repeated-measures ANOVA that includes Trial as a within-subject factor to examine task effects. Given the trial-level structure of the data and the clear nesting of trials within participants, it would be helpful for the authors to clarify why a repeated-measures ANOVA was chosen over a linear mixed-effects model (LMM), which is commonly recommended for trial-level behavioral data. If the authors consider the repeated-measures ANOVA to be more appropriate in this context, a brief justification—ideally supported by relevant methodological references—would strengthen the transparency and interpretability of the analysis.

10. Multiple correlational analyses and control for multiple comparisons.

The manuscript reports several correlational analyses linking learning and attribution bias measures with depressive symptoms and self-esteem. However, no correction for multiple comparisons appears to have been applied. Given that multiple correlations were tested, it is possible that some of the reported effects would not remain significant under standard correction procedures. The authors are therefore encouraged to clarify whether corrections for multiple testing were considered, and if not, to explicitly justify this decision (e.g., based on preregistration, confirmatory versus exploratory status, or theoretical constraints). Alternatively, reporting adjusted p-values or clearly labeling these analyses as exploratory would improve statistical transparency.

Discussion

11. The opening paragraph does not clearly summarize the main goals and core findings of the study. It is difficult to discern from the first paragraph which specific problem it resolves and what the primary contribution of the manuscript is. For example, the discussion briefly notes the replication of prior findings, yet replication does not appear to be a central objective outlined in the Introduction and may be more appropriately addressed later in the Discussion. Similarly, the emphasis on “trial-by-trial” effects is introduced early on without a clear connection to the main research questions, which may obscure rather than clarify the study’s primary contribution.

12. Several inferential statements appear stronger than warranted by the data. In particular, model-based results are described using strong causal language (e.g., “causal attributions directly influence the dynamics of trial-by-trial self-belief formation in real-time”). While computational modeling suggests a modulation of belief updating by attributional judgments, the discussion would benefit from more clearly distinguishing between model-based inference and direct psychological causation.

13. The discussion introduces the idea of two biases—negativity bias in learning and self-serving attributional bias—and emphasizes their “concurrent occurrence.” However, it is not sufficiently clear what is meant by this term. It remains

ambiguous whether “concurrent occurrence” simply refers to the presence of both biases within the same task, or whether a functional or mechanistic relationship between the two is implied. Clarifying the intended meaning and theoretical significance of their co-occurrence would strengthen this section.

14. The statement “belief formation was also attenuated in the interference condition with better-than-expected feedback” does not appear to be supported by a direct statistical test. Rather, this conclusion seems to be inferred from the structure of the computational model, in which attribution-related down-weighting applies equally to both positive and negative prediction errors. Clarifying this distinction, or explicitly reporting analyses that directly test this claim, would substantially improve the interpretability and precision of the discussion.

Reviewer #2 (Remarks to the Author):

1. Are the reported effects specific to self related learning? The current design lacks a control condition involving learning about others, which raises the possibility that the observed effects reflect general reward/punishment processing rather than self relevant processing.
2. The Introduction states that individuals typically show an optimistic update bias, whereas the present study reports a negative bias. This inconsistency with prior findings needs to be discussed more thoroughly.
3. The interaction effects in the behavioral results need to be explained more clearly. At present, it is not clear which specific behavioral effects the computational model is intended to explain. There appears to be a mismatch between the behavioral and modeling analyses, as they capture different aspects of behavior. The behavioral analyses should be structured in a way that better corresponds to the model based analyses.
4. It is unclear whether trials in which participants made external attributions were also trials with more extreme feedback. This raises two concerns:
 - (1) In the modeling analysis, the effects attributed to attribution weights and feedback weights may partially overlap.
 - (2) Differences in the proportion of external attributions under positive versus negative prediction errors may be driven by differences in feedback extremity rather than attribution valence.
5. The correlational results are relatively weak and raise concerns about multiple comparison correction. In addition, it is unclear whether the sample size provides sufficient statistical power for individual difference analyses. Overall, the reliability of these results should be discussed.
6. The authors note that the current sample shows predominantly subclinical symptom levels. Why is this the case? Was the sample selectively recruited? This characteristic may be related to the observed negativity bias and should be discussed.
7. How was the sample size determined?
8. The statement that “two participants were excluded due to missing variance in their responses” requires clarification. What exactly is meant by “missing variance”?
9. The manuscript states that trials from all conditions were intermixed in a fixed order. Why was a fixed order used, and how was this order determined?
10. The feedback was determined by a series of fixed prediction errors relative to participants' current beliefs. The exact procedure for generating and implementing these prediction errors needs to be described in detail.
11. The influence of relative probability density on feedback values was modeled as linear. Have alternative, potentially nonlinear mappings been considered?
12. Typo: “On average, participants showed higher learning rates for negative ($Mdn = .024$)” should be $Mdn = .24$.

Communications Psychology is committed to improving transparency in authorship. As part of our efforts in this direction, we are now requesting that all authors identified as ‘corresponding author’ create and link their Open Researcher and Contributor Identifier (ORCID) with their account on the Manuscript Tracking System prior to acceptance. ORCID helps the scientific community achieve unambiguous attribution of all scholarly contributions. You can create and link your ORCID from the home page of the Manuscript Tracking System by clicking on ‘Modify my Springer Nature account’ and following the instructions in the link below. Please also inform all co-authors that they can add their ORCIDs to their accounts and that they must do so prior to acceptance.

If you experience problems in linking your ORCID, please contact the [Platform](http://platformsupport.nature.com/)

Support Helpdesk.

Version 1:

Decision Letter:

Dear Dr Mayer,

Your manuscript titled "Causal attributions shape the formation of novel self-beliefs" has now been seen by our reviewers, whose comments appear below. In light of their advice I am delighted to say that we are happy, in principle, to publish a suitably revised version in Communications Psychology.

We therefore invite you to revise your paper one last time to address the remaining concerns of our reviewers and a list of editorial requests. At the same time we ask that you edit your manuscript to comply with our format requirements and to maximise the accessibility and therefore the impact of your work.

EDITORIAL REQUESTS:

SUBMISSION INFORMATION:

OPEN ACCESS:

*** TRANSPARENT PEER REVIEW:** Communications Psychology uses a transparent peer review system. On author request, confidential information and data can be removed from the published reviewer reports and rebuttal letters prior to publication. If you are concerned about the release of confidential data, please let us know specifically what information you would like to have removed. Please note that we cannot incorporate redactions for any other reasons.

*** CODE AVAILABILITY:** All Communications Psychology manuscripts must include a section titled "Code Availability" at the end of the methods section. We require that the custom analysis code supporting your conclusions is made available in a publicly accessible repository at this stage; please choose a repository that generates a digital object identifier (DOI) for the code; the link to the repository and the DOI must be included in the Code Availability statement. Publication as Supplementary Information will not suffice.

*** DATA AVAILABILITY:**

All Communications Psychology manuscripts must include a section titled "Data Availability" at the end of the Methods section. More information on this policy, is available in the Editorial Requests Table and at <http://www.nature.com/authors/policies/data/data-availability-statements-data-citations.pdf>

Link Redacted

Best regards,

Jennifer Bellingtier

Jennifer Bellingtier, PhD
Senior Editor
Communications Psychology

Hu Chuan-Peng, PhD
Editorial Board Member
Communications Psychology
orcid.org/0000-0002-7503-5131

REVIEWER EXPERTISE:

Self-concept; self-belief, depression, computational modelling

REVIEWERS' COMMENTS:

Reviewer #1 (Remarks to the Author):

Thank you to the authors for the substantial revisions. The manuscript has clearly improved in several important respects. In particular, I appreciate the added detail regarding the cover story and debriefing, the replacement of the repeated-measures ANOVA with a trial-level linear mixed-effects model, the revised attribution analyses controlling for feedback extremity, and the more transparent treatment of multiple comparisons and power considerations. These revisions address a number of my earlier methodological concerns and substantially improve the manuscript's transparency and rigor. I believe the manuscript is now considerably stronger than in the previous round. My remaining comments are intended mainly to help further tighten the conceptual framing and ensure that the interpretation stays closely aligned with the current evidence.

1. Although the authors have softened some of the causal language and added a useful limitations section, I still think that the overall scope of the Discussion and Conclusion remains somewhat broader than the present data can directly support. As currently analyzed, the study most directly shows a negativity bias in self-related learning; associations of learning bias scores with depressive symptoms and self-esteem; and an effect of prediction error valence on external attributions in both attribution models. In addition, depressive symptoms were associated with lower external attributions overall, whereas self-esteem showed a significant interaction with prediction error valence. However, the manuscript continues to move toward broader claims about "self-belief formation," "maintenance" of self-beliefs, and "enduring negative self-beliefs". In particular, "formation" does not appear to have been operationalized as a distinct outcome beyond belief updating, and "maintenance" would seem to require stronger longitudinal or recursive evidence. Given that the sample consists of undiagnosed students and that the authors themselves acknowledge the lack of directional or causal inference, I would encourage a further narrowing of the mechanistic and clinical claims so that they more closely match the actual evidential scope of the study. Relatedly, the final sentence of the abstract may still overstate the scope of the findings. The sample consists of individuals from a non-clinical student population with varying levels of depressive symptoms, rather than patients with a clinical diagnosis of depression. Formulations referring directly to effects "in depression" therefore risk exceeding what the present data can support. A more cautious formulation referring to depressive symptom severity or tendency in a non-clinical sample would be more appropriate.

2. The term "self-belief" may be too broad for the construct actually measured in this study. Based on the task and outcome measures, the study appears to assess updating of performance- or ability-related expectations, rather than self-beliefs in a broader sense. More specific language, such as "self-related belief updating," "beliefs about one's own ability," or "performance-related expectations/confidence," may better reflect the construct actually assessed. The authors may also find it useful to consider recent work distinguishing confidence from broader self-beliefs (Hoven, M., Luigjes, J., Denys, D. et al. How do confidence and self-beliefs relate in psychopathology: a transdiagnostic approach. *Nat. Mental Health* 1, 337–345 (2023). <https://doi.org/10.1038/s44220-023-00062-8>).

3. It is also unclear why self-esteem is not discussed alongside depression in the third hypotheses section (We also anticipated that with increasing depressive symptom severity individuals would more likely attribute failures, defined as worse-than-expected feedback, to internal causes, while attributing successes, defined as better-than-expected feedback, to external causes, as implicated by the attributional theory of depression), given that both constructs are examined throughout the manuscript.

4. It is also notable that the concept of "self-serving bias" is introduced for the first time in the Results section rather than in the Introduction.

5. Where interactions are reported as significant, it would be helpful to provide more explicit follow-up analyses or a clearer explanation of their direction and form. At present, some interaction effects are noted as significant, but the substantive

pattern they represent remains somewhat underexplained.

6. The Discussion does acknowledge that depressive symptoms and self-esteem yielded different attributional results, which I appreciate. I also value the authors' attempt to explain this asymmetry. However, the current explanation could be strengthened by a more explicit theoretical account, for example, by clarifying the conceptual and mechanistic differences between depression and self-esteem. These patterns may instead reflect different psychological processes and carry different theoretical implications. Developing this point further would, in my view, substantially strengthen the theoretical contribution of the Discussion.

7. Phrases such as "distinct attributional patterns and biased learning" may overstate what the present attribution analyses can support, because depressive symptoms and self-esteem were modeled separately and not directly compared. As such, the current results do not seem sufficient to conclude that they reflect distinct attributional patterns.

8. The phrasing "with increasing depressive symptom severity individuals" can be more clearly formulated, such as "individuals with more severe depressive symptoms" or "greater depressive symptom severity was associated with...."

Reviewer #2 (Remarks to the Author):

The authors have addressed all of my concerns.

Response to reviewers

We sincerely thank both reviewers for their thoughtful and constructive feedback. Their comments have helped us to substantially improve the clarity, rigor, and overall quality of the manuscript.

In response to concerns raised by both reviewers regarding the robustness of our findings on individual differences and their associations with attribution and learning biases, we have taken a more cautious interpretational approach. We have revised the title of the manuscript from “*Self-esteem modulates beneficial causal attributions in the formation of novel self-beliefs*” to “*Causal attributions shape the formation of novel self-beliefs*.” This change better reflects the strength of the evidence presented and avoids overstating conclusions about individual differences. Additionally, during this round of revision, we identified and corrected an error related to participant exclusion. After implementing the correction, we re-ran all relevant analyses. Importantly, this did not affect the overall pattern of results or the conclusions drawn. However, some statistical values have changed slightly, and these updates are now accurately reflected throughout the manuscript.

We are grateful for the reviewers’ careful reading and valuable suggestions. Please find below a point-by-point response to all comments.

Reviewer #1

1. The articulation of the theoretical gap requires clarification. The manuscript notes that prior work has relied largely on self-report measures and that less is known about real-time causal attributions during self-related learning. This is a potentially strong and important gap; however, it is not formulated sharply enough. In particular, the contrast between what is known (e.g., attributional styles assessed retrospectively or at a trait level) and what remains unknown (e.g., trial-by-trial causal attributions interacting with belief updating) should be made more explicit and systematic.

Thank you for this helpful suggestion. In response, we substantially revised the section to articulate the theoretical gap more explicitly. In the new paragraph, we clarify two specific gaps in the literature: 1. the largely unexplored question of how causal attributions made in real time dynamically shape belief updating as it unfolds, 2. the relative lack of work examining how such attributional processes operate in self-related learning (compared to other-/world-related learning), particularly regarding beliefs about one’s own abilities.

“Taken together, this body of work highlights the importance of causal attributions in shaping belief formation. However, it also leaves two important gaps. First, most prior research has conceptualized attributional style primarily as a stable interindividual trait (Elig & Frieze, 1979), rather than examining how causal attributions made in real time dynamically influence belief updating as it unfolds. Second, although recent computational approaches have incorporated trial-by-trial attributions (Dorfman et al., 2019, 2021; Zamfir & Dayan, 2022), their focus has largely been on learning about action–outcome contingencies in the external environment. Considerably less attention has been paid to how these attributional processes shape learning about

one's own abilities. Addressing these gaps is essential for understanding how moment-to-moment causal interpretations contribute to the formation and maintenance of self-related beliefs.”

2. The flow of the Introduction is weakened by redundancy and shifts in focus. Statements regarding negative biases in self-related learning in depression and low self-esteem appear both before and after the research questions are introduced, which disrupts the logical progression. Relatedly, some literature review elements appear after the explicit statement of research aims, whereas they would more naturally precede and motivate these aims.

We agree that the original version of the Introduction contained some shifts in focus that may have disrupted the logical flow. In response, we substantially reorganized the structure of the Introduction to ensure a clearer and more coherent progression. The revised structure now proceeds as follows: (1) attributional styles and their relationship with mental health, particularly depression; (2) negatively biased self-related learning and belief formation in depression and low self-esteem; (3) prior research on the relationship between causal attributions and belief formation; (4) identification of the key gaps in the literature; and finally, (5) the research questions, hypotheses, and a brief overview of the methods.

3. The research questions themselves are relevant and clearly stated, but they are introduced somewhat abruptly. The Introduction would benefit from a more gradual build-up toward these questions, ensuring that the reader clearly understands why these questions logically follow from the preceding theoretical discussion.

Thank you for this helpful comment. We aimed to solve this issue by moving the research questions more towards the end of the Introduction, where they are more clearly motivated by previous research (see response to comment #2).

Methodological Clarifications and Requests

4. Clarification of the cover story and agent manipulation

The following points relate to the transparency and validity of the cover story and feedback manipulation.

The manuscript refers to a “cover story” involving possible manipulation of feedback by a computer agent, but the details of this cover story are not sufficiently described. It remains unclear how the possibility of computer interference was explained to participants, why such manipulation was introduced, and how it was justified within the task narrative. Providing a clearer description of the cover story is important for evaluating participants’ understanding of the task and the validity of the attribution manipulation.

Thank you for raising this important point. In the experiment, the cover story was introduced as part of a study on “cognitive estimation ability”. Participants were informed that they would answer estimation questions across different categories and receive trial-by-trial feedback on their performance. To operationalize the attribution manipulation, participants were explicitly told that in some trials the computer agent might alter the feedback they received. This manipulation was framed as a feature of the experimental environment, and participants were instructed that their task was not only to perform the estimation but also to judge

whether the feedback reflected their true performance or had been modified by the computer. Apart from that, the participants were not given any further information about the purpose of this potential feedback manipulation or other alleged study hypotheses.

More specifically, participants read the following instructions:

“This study examines cognitive estimation ability. You will soon be presented with estimation questions in various categories. These estimation categories are comparable in terms of average difficulty, but individuals may differ (sometimes significantly) in their estimation performance. The goal is to answer these questions as accurately as possible.

In addition to answering the estimation tasks, another goal is to predict your own estimation performance as accurately as possible. After each trial, you will receive feedback on how well you performed on the task. In some trials, the computer may manipulate your feedback, meaning you might see feedback indicating a better or worse performance than you actually had. Your task in these trials is to determine whether the feedback reflects your actual performance or has been manipulated. In each trial, you will be informed whether it is possible that the computer has manipulated the feedback.”

Thus, the key element of the cover story was the suggestion that feedback reflected (at least in part) their true estimation ability, whereas in reality it was entirely experimentally controlled. The rationale for this design was to investigate how individuals form beliefs about their own ability and learn from feedback under conditions of uncertainty about its origin. By instructing participants to both provide estimates and predict their own performance, we aimed to ensure that they engaged with the task in a way that made the feedback personally meaningful. At the same time, the introduction of a potential external source of distortion (i.e., manipulation by the computer) created controlled ambiguity regarding feedback validity, which is central to our research question on attribution.

We acknowledge that the manuscript provided little detail on how the cover story was implemented. We have revised the manuscript to more clearly describe the cover story and explicitly state the nature of the deception regarding feedback:

“As part of the cover story, the LOOP task was presented as a study on cognitive estimation ability, with feedback described as reflecting participants’ performance. In reality, all feedback was pre-programmed and independent of actual responses in the estimation task. Two categories were randomly assigned to predominantly better-than-expected feedback (High Ability condition), and two to predominantly worse-than-expected feedback (Low Ability condition). Participants were informed that in two categories (Agent condition) the computer might alter the feedback, such that performance could be either improved in one category or worsened in the other. The frequency and magnitude of these potential manipulations were not specified (see Supplementary Materials for full instructions). In these trials, in addition to providing estimates, participants were asked to judge whether the feedback reflected their true performance or had been manipulated. Importantly, the feedback in the Agent

condition was pre-programmed in the same way as in the corresponding categories without supposed manipulation (No Agent condition).”

We also provide additional information on what the debriefing process entailed:

“During debriefing, participants were informed that the feedback had been pre-programmed and did not reflect their actual performance, and the purpose of the manipulation was explained.”

Further, we added the full instruction text to the Supplemental Materials.

5. Belief in the cover story and participant exclusion

Previous work using similar paradigms (e.g., studies by Laura Müller-Pinzler et al.) has reported that some participants did not believe the cover story and were therefore excluded from analyses. It would be important to clarify whether a similar issue occurred in the present study. Specifically, were participants' beliefs in the cover story assessed, and were any participants excluded due to explicit disbelief or suspicion regarding the agent manipulation?

Yes, we assessed whether participants believed the cover story in a written follow-up survey at the end of the LOOP task (“Did you generally feel that the feedback was related to your performance?”). Additionally, after completing the study, participants were briefly surveyed about their impressions of the task by the study staff. Participants were excluded only if they explicitly stated in the written follow-up survey that they did not believe the feedback was related to their performance at any point during the task. Non-specific expressions of suspicion (e.g., “The feedback seemed a bit random to me; I was often confused by the varying quality of my estimations, even though the feedback was not manipulated.”) as well as beliefs that the feedback was consistently biased but still related to performance (e.g., “I think it [the feedback] was generally made worse to make me feel worse.”) did not lead to exclusion. This approach was chosen to avoid excluding participants based on suspicions potentially induced by the follow-up questions. Based on this criterion, no participants were excluded from the analyses.

To clarify this issue, we added information about the follow-up survey to the “Questionnaires and debriefing” section in the Methods:

“After completing the LOOP task and questionnaires, participants took part in a brief follow-up survey assessing their impressions of the task. This survey included questions about the perceived plausibility of the feedback during the task, as well as participants' retrospective estimates of how frequently the computer manipulated the feedback (Supplementary Figure S1). Although some participants expressed non-specific suspicions about the feedback, none explicitly indicated disbelief in the cover story, and no participants were excluded on this basis.”

6. Precision of performance expectation ratings

Based on the information of the figure, participants were not shown their exact numerical performance expectation ratings (e.g., percentile values) during the task. If these values were not explicitly displayed, participants' expectations may have been relatively imprecise

compared to the explicit and numerical feedback provided. Please clarify whether potential imprecision in expectation estimates could have influenced how participants perceived and interpreted discrepancies between expectations and feedback.

Thank you for pointing this out. Our figure was misleading in this respect. Our participants could in fact see the exact numerical value while they were adjusting or viewing their expectation rating. The number was displayed together with the movable cursor on the rating scale. We corrected Figure 1A accordingly.

7. Specification of feedback magnitudes and distribution across conditions

While it is understandable that the Ability condition was manipulated via predominantly positive versus predominantly negative feedback, additional detail is needed regarding the exact structure of the feedback. Specifically, how large were the deviations between feedback and expectations (i.e., the magnitude of prediction errors), and did these deviations fall within a predefined range? Furthermore, please clarify how many positive versus negative feedback trials were presented within each condition (high vs. low ability), and whether this distribution was identical across participants.

Thank you for this helpful comment. We now provide the requested details in a new paragraph within the Methods section:

“Instead of providing fixed feedback values, the feedback sequence was designed to elicit prediction errors of specific magnitudes and valences 20,21,32. Planned prediction errors were drawn from a hand-designed sequence that followed a predefined distribution for each condition (High Ability: -18 to 27, 70% positive, 30% negative; Low Ability: -27 to 18, 30% positive, 70% negative). The sequence was identical for all participants and followed a fixed order within each experimental condition. For each trial, the feedback value was calculated by adding the corresponding planned prediction error from the sequence to the participant’s current ability belief, defined as the average of their last five expectation ratings within the respective condition. Before participants had provided their first performance expectation rating, the expectation value was set to 50%. This approach resulted in varying feedback sequences across participants while keeping actual prediction errors largely independent of individual performance expectations. It also ensured a relatively balanced distribution of actual negative prediction errors (Agent: $M = -12.7$, 45.9% of trials; No Agent: $M = -12.6$, 45.4%) and positive prediction errors (Agent: $M = 14.5$, 54.1%; No Agent: $M = 14.2$, 54.6%) across conditions.”

8. Assessment of belief in agent interference during debriefing

Relatedly, the debriefing section does not report to what extent participants believed that their feedback had actually been manipulated by the agent. If participants’ beliefs in agent interference were measured explicitly (e.g., via post-task questions or ratings), reporting this information would substantially strengthen the interpretation of the attribution manipulation. If such measures were not collected, this limitation should be acknowledged.

Thank you for highlighting this issue. Apart from trial-by-trial attributions of feedback in the Agent condition, we also measured participants’ belief in agent interference in the follow-up

survey mentioned in the response to comment #5. Specifically, participants were asked to retrospectively rate the frequency of agent interference:

“Think about the trials in which it was possible for the feedback to be manipulated in a positive direction, that is, improved. Looking back, in what percentage of cases do you think the computer manipulated the feedback?”

and

“Think about the trials in which it was possible for the feedback to be manipulated negatively, that is, worsened. Looking back, in what percentage of cases do you think the computer manipulated the feedback?”

Ratings ranged between 0 and 100% for negative manipulations and 5 and 100% for positive manipulations (see Figure R1 below). We did not exclude participants based on these ratings. To clarify that we explicitly measured participants’ belief in agent interference, we added the following statement to the “Questionnaires and debriefing” section in the Methods:

“After completing the LOOP task and questionnaires, participants took part in a brief follow-up survey assessing their impressions of the task. This survey included questions about the perceived plausibility of the feedback during the task, as well as participants’ retrospective estimates of how frequently the computer manipulated the feedback (Supplementary Figure S1). Although some participants expressed non-specific suspicions about the feedback, none explicitly indicated disbelief in the cover story, and no participants were excluded on this basis.”

We further added the following figure to the Supplemental Materials.

Figure R1: Percentage of retrospectively perceived agent interference.

9. Use of repeated-measures ANOVA versus linear mixed-effects models

The manuscript reports a repeated-measures ANOVA that includes Trial as a within-subject factor to examine task effects. Given the trial-level structure of the data and the clear nesting of trials within participants, it would be helpful for the authors to clarify why a repeated-measures ANOVA was chosen over a linear mixed-effects model (LMM), which is commonly recommended for trial-level behavioral data. If the authors consider the repeated-measures ANOVA to be more appropriate in this context, a brief justification—ideally supported by relevant methodological references—would strengthen the transparency and interpretability of the analysis.

Thank you for this helpful suggestion. In response, we have replaced the repeated-measures ANOVA with a linear mixed-effects model. Specifically, we now model participants' performance expectations at the trial level using an LMM with Trial (centered), Ability condition (High vs. Low), and Interference condition (Agent vs. No Agent) as fixed effects, including all interactions, and random intercepts and random slopes for Trial across participants. Random slopes for Ability and Interference were initially considered but not included in the final model because their inclusion led to model non-conversion.

We agree that this approach is more appropriate for the hierarchical structure of the data, as it accounts for the nesting of trials within participants and allows for individual variability in both baseline performance and change over trials. Moreover, treating Trial as a continuous predictor enables a more parsimonious and theoretically meaningful characterization of temporal dynamics compared to modeling it as a categorical factor with multiple levels, as in the repeated-measures ANOVA. We believe it improves both the transparency and interpretability of the results. The manuscript has been updated accordingly.

Methods:

“A model-agnostic analysis of participants' performance expectations was conducted to illustrate basic task effects in our behavioral data. To this end, we fit a linear mixed-effects model predicting performance expectations from Trial (centered), Ability condition (High vs. Low), and Interference condition (Agent vs. No Agent), including all interactions. The model included random intercepts and random slopes for Trial across participants.”

Results:

“A linear mixed-effects model predicting performance expectations from Trial, Interference condition and Ability condition revealed a small but significant decrease in performance expectations across trials ($\beta = -0.2$, 95% CI [-0.35, -0.04], $t(63) = -2.47$, $p = 0.016$). Across trials and Ability conditions, expectations were lower in the Agent condition compared to the No-Agent condition ($\beta = -1.45$ [-2.08, -0.82], $t(4986) = -4.53$, $p < .001$). Expectations were substantially higher for the High Ability compared to the Low Ability condition ($\beta = 5.8$ [5.17, 6.43], $t(4986) = 18.11$, $p < .001$). The interaction between Trial and Interference was not significant ($\beta = -0.02$ [-0.13, 0.09], $t(4986) = -0.31$, $p = 0.753$), suggesting that expectation change was

comparable between the Agent and No-Agent conditions irrespective of feedback valence. In contrast, the interaction between Trial and Ability was significant ($\beta = 0.31$ [0.2, 0.42], $t(4986) = 5.55$, $p < .001$), indicating that the change in performance expectations across trials differed between the High and Low Ability conditions. There was also a significant interaction between Interference and Ability ($\beta = -8.83$ [-10.09, -7.58], $t(4986) = -13.79$, $p < .001$).

Importantly, these effects were further qualified by a significant three-way interaction between Trial, Interference, and Ability ($\beta = -0.52$ [-0.74, -0.3], $t(4986) = -4.67$, $p < .001$). This interaction indicates that belief updating over time differed depending on both the valence of feedback and the attributional context (Figure 1C). In particular, participants adjusted their expectations differently in the High versus Low Ability conditions, reflecting differences in the distribution of positive and negative outcomes, and this effect depended on whether negative outcomes could be attributed to an external agent. This pattern suggests that expectation updating is shaped by valence-dependent prediction errors and modulated by attributional processes, providing the basis for the computational modeling approach.”

10. Multiple correlational analyses and control for multiple comparisons.

The manuscript reports several correlational analyses linking learning and attribution bias measures with depressive symptoms and self-esteem. However, no correction for multiple comparisons appears to have been applied. Given that multiple correlations were tested, it is possible that some of the reported effects would not remain significant under standard correction procedures. The authors are therefore encouraged to clarify whether corrections for multiple testing were considered, and if not, to explicitly justify this decision (e.g., based on preregistration, confirmatory versus exploratory status, or theoretical constraints). Alternatively, reporting adjusted p-values or clearly labeling these analyses as exploratory would improve statistical transparency.

Thank you for raising this important point regarding multiple comparisons. In response, we have revised our analytical approach in two ways. First, following an additional suggestion by a second reviewer, we restructured the analysis of attributional patterns. Rather than relying on correlational analyses with derived bias scores, we now use linear mixed-effects models that directly model attributions while controlling for feedback extremity. This approach provides a more robust and statistically appropriate test of attributional bias at the trial level.

Second, for the remaining correlational analyses (linking the learning bias with depressive symptoms and self-esteem), we now control for multiple comparisons using the false discovery rate (FDR) procedure. Adjusted p-values are reported in the revised manuscript.

In addition, for the LMM analyses, we applied FDR correction to the fixed effects of interest by defining appropriate families of tests (e.g., grouping p-values associated with the main effects of prediction error valence and individual difference variables such as self-esteem or depressive symptoms). This ensures a consistent and transparent approach to controlling for multiple testing across all key inferential results.

These changes have been implemented in the manuscript, and we believe they substantially improve the statistical rigor and transparency of our analyses.

Methods:

“Spearman correlations were then calculated between individual bias scores and PHQ-9 and SDQ-III sum scores, respectively. To account for multiple hypothesis testing, p-values were adjusted using the Benjamini-Hochberg false discovery rate (FDR) procedure.”

“To quantify a potential attributional bias in the Agent condition – the tendency to attribute better-than-expected feedback to one’s own ability while attributing worse-than-expected feedback to manipulation by the computer agent – we tested whether attribution patterns varied as a function of prediction error valence. Because feedback values closer to the ends of the feedback scale may be perceived as less plausible and could therefore increase external attributions independent of prediction error valence, feedback extremity was included as a control predictor.

To this end, we fit two linear mixed-effects models predicting the proportion of external attributions. Both models included feedback extremity decomposed into within-subject and between-subject components to distinguish trial-by-trial fluctuations in feedback extremity from individual differences in average feedback extremity. Prediction error valence and participants’ depressive symptoms or self-esteem (mean-centered) were included as predictors, as well as their interaction. Due to the high correlation between depression and self-esteem, these variables were entered in separate models to avoid multicollinearity. Both models included random intercepts for participants and random slopes for prediction error valence by participant. Random slopes for feedback extremity were initially considered but were not included in the final model because their inclusion led to model non-convergence. For each fixed effect of interest, p-values obtained from the two linear mixed-effects models were jointly adjusted for multiple comparisons using the FDR-procedure, such that corresponding effects (e.g., main effect of prediction error valence, interaction with depression/self-esteem) were corrected together.”

Results:

“To assess the relationship of depressive symptoms and self-esteem with a valence bias in learning, we calculated individual bias scores for each participant. Correlation analyses showed a negative association between bias scores and depressive symptoms (Spearman $\rho = -0.26$, 95% CI [-0.47, -0.01], $p_{\text{FDR}} = 0.04$) as well as a positive association between bias scores and self-esteem ($\rho = 0.44$ [0.21, 0.62], $p_{\text{FDR}} < .001$, Figure 3D). This suggests that individuals with elevated depressive symptoms and lower self-esteem showed a stronger negativity bias in learning. Depression and self-esteem were strongly negatively correlated ($\rho = -0.62$ [-0.75, -0.45], $p_{\text{FDR}} < .001$).”

“We examined how the proportion of external attributions varied with prediction error valence and participants’ depressive symptoms and self-esteem, controlling for feedback extremity. To this end, we fit two separate linear mixed-effects models that

were identical except that one included depression and the other included self-esteem as a predictor. The results of both models are shown in Figure 4.

[...]

Prediction error valence significantly predicted external attributions in both models, with worse-than-expected feedback associated with more external attributions (depression model: $\beta = -0.0547$ [-0.1024, -0.0066], $t(70) = -2.23$, $p_{FDR} = 0.029$; self-esteem model: $\beta = -0.0547$ [-0.101, -0.0082], $t(71) = -2.31$, $p_{FDR} = 0.029$). That is, participants were more likely to attribute feedback to a manipulation by the computer agent when the feedback was worse than expected than when it was better than expected (see Figure 3C for raw data). This corresponds to a self-serving attributional bias.

Depressive symptoms were negatively associated with external attributions ($\beta = -0.0075$ [-0.0129, -0.002], $t(63) = -2.67$, $p_{FDR} = 0.019$), but the interaction between prediction error valence and depressive symptoms was not significant ($\beta = 0.0045$ [-0.0034, 0.0124], $t(63) = 1.11$, $p_{FDR} = 0.269$). In contrast, self-esteem alone was not significantly associated with external attributions ($\beta = 0.0011$ [-0.0039, 0.0062], $t(65) = 0.44$, $p_{FDR} = 0.665$), but the interaction between prediction error valence and self-esteem was significant ($\beta = -0.0078$ [-0.0142, -0.0013], $t(68) = -2.34$, $p_{FDR} = 0.045$). This indicates that the effect of prediction error valence on external attributions was stronger among participants with higher self-esteem; in other words, participants with higher self-esteem showed a stronger self-serving bias (Figure 4B).”

Discussion

11. The opening paragraph does not clearly summarize the main goals and core findings of the study. It is difficult to discern from the first paragraph which specific problem it resolves and what the primary contribution of the manuscript is. For example, the discussion briefly notes the replication of prior findings, yet replication does not appear to be a central objective outlined in the Introduction and may be more appropriately addressed later in the Discussion. Similarly, the emphasis on “trial-by-trial” effects is introduced early on without a clear connection to the main research questions, which may obscure rather than clarify the study’s primary contribution.

We appreciate this helpful comment and agree that the opening paragraph lacked a clear connection to the research questions presented in the Introduction. We have revised the section to more clearly articulate the study’s main findings:

“The present study examined how momentary causal attributions shape the formation of self-beliefs. Specifically, we assessed whether attributions of feedback in a validated learning task influence belief updating, and whether individual differences in self-esteem and depressive symptoms are associated with distinct attributional patterns and biased learning. Moving beyond trait-like conceptualizations of attributional style, this approach captures how causal interpretations dynamically unfold and shape learning about one’s own abilities in real time. Our findings provide three main insights. First, using computational modeling, we show that momentary causal attributions systematically modulated self-related learning. When feedback was attributed to an external agent, belief updating was attenuated, indicating that participants adjusted their self-beliefs to a lesser degree when they thought that the

feedback had been manipulated. Second, we observed a negativity bias in self-related learning, showing that participants adjusted their self-beliefs to a stronger extent after worse-than-expected feedback, compared to better-than-expected feedback. This negativity bias was more pronounced in individuals with higher levels of depression and lower self-esteem. Third, our results suggested a self-serving attributional pattern, such that worse-than-expected feedback was more likely attributed to external causes, whereas better-than-expected outcomes were attributed to one's own ability. Less self-serving attributional tendencies were specifically associated with lower self-esteem, rather than depressive symptom severity."

12. Several inferential statements appear stronger than warranted by the data. In particular, model-based results are described using strong causal language (e.g., "causal attributions directly influence the dynamics of trial-by-trial self-belief formation in real-time"). While computational modeling suggests a modulation of belief updating by attributional judgments, the discussion would benefit from more clearly distinguishing between model-based inference and direct psychological causation.

Thank you for pointing this out. We have carefully revised the manuscript to more clearly distinguish between model-based inferences and claims about psychological causation. Specifically, we have softened or removed causal language throughout (e.g., replacing formulations such as "directly influenced" with "modulated by", "associated with"). Further, we explicitly clarified where conclusions are derived from the structure of the winning computational model rather than from direct statistical tests (see also responses to comments #13 and 14). We have also added a dedicated limitations section in which we emphasize that our findings, particularly associations involving self-esteem and depressive symptoms, are correlational and do not permit conclusions about causal mechanisms or recursive effects:

"Third, given that depressive symptoms and self-esteem were not experimentally manipulated, the directionality and causality of their associations with attributional patterns remain unclear. Theories assume recursive dynamics between aspects of the self-concept and attributional patterns (Bandura, 1986, 1997; Marsh, 1990; Silver et al., 1995; Stajkovic & Sommer, 2000). While our current analysis demonstrates self-serving attributional patterns and shows how these may influence belief updating over time, it does not allow conclusions about the causal interplay between the resulting beliefs and self-esteem or depressive symptoms, nor about whether such beliefs subsequently shape future attributions."

13. The discussion introduces the idea of two biases—negativity bias in learning and self-serving attributional bias—and emphasizes their "concurrent occurrence." However, it is not sufficiently clear what is meant by this term. It remains ambiguous whether "concurrent occurrence" simply refers to the presence of both biases within the same task, or whether a functional or mechanistic relationship between the two is implied. Clarifying the intended meaning and theoretical significance of their co-occurrence would strengthen this section.

Thank you for this comment. We agree that "concurrent occurrence" is a vague description and could be read as implying some kind of interaction or causal linkage. However, this is

not what we want to express, since we did not analyse relationships between these two sources of bias. In the revised document, we now explicitly state the co-occurrence of these biases without implying a relationship:

“Although both the negativity bias (Czekalla et al., 2024; Müller-Pinzler et al., 2019, 2022; Schröder et al., 2025) and the self-serving attributional bias (Campbell & Sedikides, 1999; Dorfman et al., 2019, 2021) have been documented independently in healthy populations, our results indicate that they can both affect belief formation within the same context, while remaining analytically separable.”

We have also revised the following paragraph to more explicitly tie each bias to the relevant task condition and separate their contributions:

“In our task, the possibility of external interference specifically introduced uncertainty about the validity of the feedback in the Agent condition. Under these circumstances, participants may have used external attributions as a cognitive safeguard, treating outcomes as potentially not fully reflective of their own performance. As implied by the structure of our winning computational model, this attributional tendency selectively attenuated the integration of feedback, resulting in relatively stable performance expectations in the Agent condition. This effect was more pronounced for worse-than-expected outcomes, which were more likely to be attributed to external causes under the self-serving attributional bias. In contrast, across all conditions, we observed a general negativity bias in learning, whereby worse-than-expected feedback exerted a stronger influence on belief updating than better-than-expected feedback. Thus, while the negativity bias shaped belief updating overall, external attributions in the Agent condition reduced the extent to which such negative feedback was incorporated. Taken together, in contexts with ambiguous sources of self-related feedback, the attributional bias may support the maintenance of desirable self-beliefs by strengthening self-efficacy and self-coherence (Mezulis et al., 2004; Silver et al., 1995; Stajkovic & Sommer, 2000). In parallel, the negativity bias in learning may serve to guard against overly optimistic or potentially erroneous belief updates.”

14. The statement “belief formation was also attenuated in the interference condition with better-than-expected feedback” does not appear to be supported by a direct statistical test. Rather, this conclusion seems to be inferred from the structure of the computational model, in which attribution-related down-weighting applies equally to both positive and negative prediction errors. Clarifying this distinction, or explicitly reporting analyses that directly test this claim, would substantially improve the interpretability and precision of the discussion.

Thank you for highlighting this issue. As outlined above, we have now revised the entire paragraph to more clearly separate the contributions of each bias. In this new paragraph, we now explicitly state that the influence of attributional biases on belief formation is implied by our model instead of supported by a statistical test:

“As implied by the structure of our winning computational model, this attributional tendency selectively attenuated the integration of feedback, resulting in relatively stable performance expectations in the Agent condition.”

Reviewer #2

1. Are the reported effects specific to self related learning? The current design lacks a control condition involving learning about others, which raises the possibility that the observed effects reflect general reward/punishment processing rather than self relevant processing.

Thank you for raising this important point. We agree that, because the current experimental design does not include a control condition involving learning about others, we cannot directly test whether the reported effects are specific to self-relevant processing within this dataset.

At the same time, we note that in several of our previous studies using the LOOP task (e.g., (Czekalla et al., 2024; Müller-Pinzler et al., 2019; Schröder et al., 2025), we included a condition in which participants learned about others' performance. Across these studies, we consistently found a negativity bias that was only observed when participants learned about their own abilities. We thus have evidence for a self-specific negativity bias using the LOOP task. In the present study, we therefore chose to drop the "Other" condition in order to accommodate the additional experimental manipulation (i.e., the interference condition) without substantially increasing task duration and participant burden. Further, there is evidence that suggests that self-serving attributional tendencies are specifically present when evaluating outcomes of one's own actions compared to others' actions, especially in competitive contexts (Malle, 2006; Zuckerman, 1979). Thus, we believe that the two biases that we report in the current study are tied to self-referential contexts.

We however think that our winning computational model captures a more general process, namely, reduced updating from outcomes perceived as externally caused, since this has also been observed in non-self-referential learning contexts (Dorfman et al., 2019, 2021). As such, while the learning bias and attributional bias are likely self-specific, the underlying computational mechanism is not necessarily specific to self-related processing. Our findings therefore extend previous evidence on causal attributions in learning to contexts that involve self-relevant information. We now acknowledge this in the Discussion section:

"Notably, while both biases are likely tied to self-referential processing (Czekalla et al., 2024; Malle, 2006; Müller-Pinzler et al., 2019; Schröder et al., 2025; Zuckerman, 1979), the computational mechanism captured by our model – reduced updating from outcomes perceived as externally caused – may reflect a more general learning principle that is not limited to self-related contexts."

2. The Introduction states that individuals typically show an optimistic update bias, whereas the present study reports a negative bias. This inconsistency with prior findings needs to be discussed more thoroughly.

Thank you for this helpful comment. We agree that the original phrasing in the Introduction may have led to confusion regarding the apparent inconsistency between an optimistic update bias and the negative bias reported in the present study.

To clarify this issue, we have revised the Introduction to more explicitly distinguish between two well-established but distinct domains of belief updating. Specifically, prior work has shown that (i) healthy individuals tend to exhibit an optimistic update bias when estimating the likelihood of future events, whereas (ii) learning about self-referential characteristics, such as one's own abilities, is often characterized by a negativity bias. These effects reflect domain-specific differences in belief updating rather than contradictory findings. In individuals with low self-esteem and clinical depression, evidence suggests that both of these biases are shifted in a negative way, meaning that the optimism bias is reduced or completely absent, and the negativity bias is more pronounced.

In the revised manuscript, we now clearly separate these two lines of research and explicitly state that the present study pertains to learning about own abilities, for which a negativity bias is expected. We hope that this clarification resolves the potential misunderstanding and improves the overall clarity of the Introduction:

“Further, within self-referential learning, healthy individuals show a negativity bias when updating beliefs about their own characteristics and abilities (Brotzeller & Gollwitzer, 2025; Müller-Pinzler et al., 2019), which is exacerbated in individuals with low self-esteem (Müller-Pinzler et al., 2019, 2022) and clinical depression (Czekalla et al., 2024). A separate line of work shows that healthy individuals tend to update their expectations about future events in an optimistically biased way (Sharot, 2011). In contrast, individuals with depression show a reduced or absent optimistic update bias (Garrett et al., 2014; Korn et al., 2014), a pattern that has also been observed in the context of social evaluation (Hoffmann et al., 2024).”

3. The interaction effects in the behavioral results need to be explained more clearly. At present, it is not clear which specific behavioral effects the computational model is intended to explain. There appears to be a mismatch between the behavioral and modeling analyses, as they capture different aspects of behavior. The behavioral analyses should be structured in a way that better corresponds to the model based analyses.

Thank you for this helpful comment. We agree that the relationship between our model-agnostic and model-based analyses was not made explicit and that this is a potential source of confusion to the reader.

Our computational model aims to describe the three-way interaction between Trial, Ability, and Interference condition. Specifically, it assumes that participants update their expectations trial-by-trial based on prediction errors, which differ systematically between the High and Low Ability conditions (predominantly positive vs. negative). In addition, the model incorporates attributional processes, such that negative outcomes can be partially attributed to an external source in the Agent condition. Thus, the three experimental factors directly map onto key model components: Trial captures expectation change over the course of the experiment, Ability condition determines the valence distribution of prediction errors, and Interference condition modulates the possibility of external versus internal attributions.

Based on a comment by another reviewer, we have revised this section entirely, since we now analyse the behavioral data using a linear mixed-effects model. We now describe the three-way interaction in more detail:

“Importantly, these effects were further qualified by a significant three-way interaction between Trial, Interference, and Ability ($\beta = -0.52 [-0.74, -0.3]$, $t(4986) = -4.67$, $p < .001$). This interaction indicates that belief updating over time differed depending on both the valence of feedback and the attributional context (Figure 1C). In particular, participants adjusted their expectations differently in the High versus Low Ability conditions, reflecting differences in the distribution of positive and negative outcomes, and this effect depended on whether negative outcomes could be attributed to an external agent. This pattern suggests that expectation updating is shaped by valence-dependent prediction errors and modulated by attributional processes, providing the basis for the computational modeling approach.”

In the following section, we now clarify that the model is intended to account for the observed three-way interaction:

“To account for the observed interaction between Trial, Ability, and Interference, we implemented a computational model of belief updating that captures how participants adjust their performance expectations over time as a function of feedback valence and attributional context.”

4. It is unclear whether trials in which participants made external attributions were also trials with more extreme feedback.

Thank you for this insightful comment. To address these concerns, we first visually inspected the relationship between external attributions and feedback extremity (see Figure R2 below). This inspection suggested that trials in which participants made external attributions were indeed associated with more extreme feedback. We also examined whether trials with negative prediction errors coincided with more extreme feedback, which they did (Figure R2).

Figure R2: Mean feedback extremity by attribution and prediction error valence. Feedback extremity was defined as the absolute difference to the middle of the feedback scale (50).

On a between-subjects level, visual inspection however indicated no association between feedback extremity, meaning that participants who received more extreme feedback overall likely did not make more external attributions (see Figure R3 below).

Figure R3: Mean feedback extremity and proportion of external attributions across both Agent conditions (Agent High, Agent Low).

This raises two concerns:

(1) In the modeling analysis, the effects attributed to attribution weights and feedback weights may partially overlap.

The above observations suggest that external attributions and extreme feedback are not fully independent at the trial level. While we cannot preclude some degree of overlap between the two parameters and their conceptual meaning, we would like to highlight several arguments that support the validity of distinguishing between these parameters in both our model and the data. First, it is important to note that attribution weights and feedback weights were estimated from different sets of trials: feedback weights were based on all trials, whereas attribution weights were derived from a subset of trials in which participants made external attributions. Although there is a tendency for more external attributions to occur for more extreme feedback values, trials with external attributions were distributed across the full range of feedback, including less extreme values. Thus, different subsets of trials contributed to the estimation of feedback weights and attribution weights.

Second, the two weighting factors are computed differently. Attribution weights implement an on/off mechanism, reducing the influence of trials with external attributions on learning, whereas feedback weights modulate the impact of trials in a continuous, nonlinear fashion, with reduced influence toward the extremes of the feedback scale. Importantly, attribution weights and feedback weights were not significantly correlated (see Supplementary Table S3), suggesting that the two parameters capture dissociable processes despite potential overlap at the trial level. This reduces the likelihood that the observed effects are solely driven by shared variance related to feedback extremity. Finally, the analyses of attributions correcting for feedback extremity (see following point (2)) further support this interpretation.

(2) Differences in the proportion of external attributions under positive versus negative prediction errors may be driven by differences in feedback extremity rather than attribution valence.

Thank you for raising this important issue. Based on the above observations, we agree that differences in the proportion of external attributions across positive versus negative prediction errors may be confounded by differences in feedback extremity. To address this concern, we revised our analysis of attributional bias. Specifically, we conducted a linear mixed-effects model predicting the proportion of external attributions while explicitly controlling for feedback extremity. To account for both within- and between-subject variability, feedback extremity was included as separate within-subject and between-subject components. This approach allowed us to isolate the effect of prediction error valence on attributional responses while accounting for potential confounding effects of feedback extremity.

In brief, the results confirmed our visual inspection and suggested that participants indeed made more external attributions after more extreme feedback. Participants who received more extreme feedback overall did not make more external attributions. However, we also found a significant effect of prediction error valence even when controlling for feedback extremity, which provides evidence for a “true” self-serving attributional bias.

The updated methods and results are now reported in the manuscript.

Methods:

“To quantify a potential attributional bias in the Agent condition – the tendency to attribute better-than-expected feedback to one’s own ability while attributing worse-than-expected feedback to manipulation by the computer agent – we tested whether attribution patterns varied as a function of prediction error valence. Because feedback values closer to the ends of the feedback scale may be perceived as less plausible and could therefore increase external attributions independent of prediction error valence, feedback extremity was included as a control predictor. To this end, we fit two linear mixed-effects models predicting the proportion of external attributions. Both models included feedback extremity decomposed into within-subject and between-subject components to distinguish trial-by-trial fluctuations in feedback extremity from individual differences in average feedback extremity. Prediction error valence and participants’ depressive symptoms or self-esteem (mean-centered) were included as predictors, as well as their interaction. Due to the high correlation between depression and self-esteem, these variables were entered in separate models to avoid multicollinearity. Both models included random intercepts for participants and random slopes for prediction error valence by participant. Random slopes for feedback extremity were initially considered but were not included in the final model because their inclusion led to model non-convergence. For each fixed effect of interest, p-values obtained from the two linear mixed-effects models were jointly adjusted for multiple comparisons using the FDR-procedure, such that corresponding effects (e.g., main effect of prediction error valence, interaction with depression/self-esteem) were corrected together.”

Results:

“We examined how the proportion of external attributions varied with prediction error valence and participants’ depressive symptoms and self-esteem, controlling for feedback extremity. To this end, we fit two separate linear mixed-effects models that were identical except that one included depression and the other included self-esteem as a predictor. The results of both models are shown in Figure 4.

Trial-by-trial feedback extremity consistently predicted external attributions across both models, with more extreme feedback associated with more external attributions (depression model: $\beta = 0.0063$, 95% CI [0.0045, 0.0081], $t(937) = 6.95$, $p < .001$; self-esteem model: $\beta = 0.0064$ [0.0045, 0.0081], $t(974) = 6.98$, $p < .001$, see Figure 4A and 4B for standardized beta coefficients). Between-subject differences in average feedback extremity were not significant in either model (depression: $\beta = 0.0018$ [-0.0045, 0.0081], $t(70) = 0.55$, $p = 0.585$; self-esteem: $\beta = 0.0000$ [-0.0068, 0.0068], $t(66) = 0.00$, $p = 0.997$).

Prediction error valence significantly predicted external attributions in both models, with worse-than-expected feedback associated with more external attributions (depression: $\beta = -0.0547$ [-0.1024, -0.0066], $t(70) = -2.23$, $p_{\text{FDR}} = 0.029$; self-esteem: $\beta = -0.0547$ [-0.101, -0.0082], $t(71) = -2.31$, $p_{\text{FDR}} = 0.029$). That is, participants were more likely to attribute feedback to a manipulation by the computer agent when the feedback was worse than expected than when it was better than expected (see Figure 3C for raw data). This corresponds to a self-serving attributional bias. Depressive symptoms were negatively associated with external attributions ($\beta = -0.0075$ [-0.0129, -0.002], $t(63) = -2.67$, $p_{\text{FDR}} = 0.019$), but the interaction between prediction error valence and depressive symptoms was not significant ($\beta = 0.0045$ [-0.0034, 0.0124], $t(63) = 1.11$, $p_{\text{FDR}} = 0.269$). In contrast, self-esteem alone was not significantly associated with external attributions ($\beta = 0.0011$ [-0.0039, 0.0062], $t(65) = 0.44$, $p_{\text{FDR}} = 0.665$), but the interaction between prediction error valence and self-esteem was significant ($\beta = -0.0078$ [-0.0142, -0.0013], $t(68) = -2.34$, $p_{\text{FDR}} = 0.045$). This indicates that the effect of prediction error valence on external attributions was stronger among participants with higher self-esteem; in other words, participants with higher self-esteem showed a stronger self-serving bias (Figure 4B).”

5. The correlational results are relatively weak and raise concerns about multiple comparison correction. In addition, it is unclear whether the sample size provides sufficient statistical power for individual difference analyses. Overall, the reliability of these results should be discussed.

Thank you for raising this important point regarding multiple comparisons. In response, we have revised our analytical approach in two ways. First, as explained in the response to the comment above, we restructured the analysis of attributional patterns. Rather than relying on correlational analyses with derived bias scores, we now use linear mixed-effects models that directly model attributions while controlling for feedback extremity. For the LMM analyses, we applied FDR correction to the fixed effects of interest by defining appropriate families of tests (e.g., grouping p-values associated with the main effects of prediction error valence and individual difference variables such as self-esteem or depressive symptoms). This

ensures a consistent and transparent approach to controlling for multiple testing across all key inferential results.

Second, for the remaining correlational analyses (linking the learning bias with depressive symptoms and self-esteem), we now control for multiple comparisons using the false discovery rate (FDR) procedure. Adjusted p-values are reported in the revised manuscript.

Methods:

“Spearman correlations were then calculated between individual bias scores and PHQ-9 and SDQ-III sum scores, respectively. To account for multiple hypothesis testing, p-values were adjusted using the Benjamini-Hochberg false discovery rate (FDR) procedure.”

Results:

“To assess the relationship of depressive symptoms and self-esteem with a valence bias in learning, we calculated individual bias scores for each participant. Correlation analyses showed a negative association between bias scores and depressive symptoms (Spearman $\rho = -0.26$, 95% CI [-0.47, -0.01], $p_{\text{FDR}} = 0.04$) as well as a positive association between bias scores and self-esteem ($\rho = 0.44$ [0.21, 0.62], $p_{\text{FDR}} < .001$, Figure 3D). This suggests that individuals with elevated depressive symptoms and lower self-esteem showed a stronger negativity bias in learning. Depression and self-esteem were strongly negatively correlated ($\rho = -0.62$ [-0.75, -0.45], $p_{\text{FDR}} < .001$).”

Further, we conducted a sensitivity analysis to determine the minimum effect size that could be reliably detected in our sample, which we report in Supplementary Note 2:

“Associations between learning bias parameters and individual differences (depression and self-esteem) were examined using Spearman rank correlations. A post hoc sensitivity analysis was conducted using G*Power, approximating the test using a bivariate correlation model. Assuming a two-tailed α level of .05 and power ($1 - \beta$) = .80, the available sample size ($N = 64$) allowed detection of correlations of at least $r = .34$. This indicates that the study was sufficiently powered to detect medium associations between individual differences in depression/self-esteem and learning biases, whereas smaller correlations may not have been reliably detected.

Associations between attributional bias and individual differences (depression and self-esteem) were now analyzed using a linear mixed-effects model. Because G*Power does not directly support mixed-effects models, the analysis was approximated using a linear multiple regression framework (R^2 increase), focusing on the unique contribution of the predictor of interest (e.g., depression or self-esteem). Assuming an alpha level of .05 and power ($1 - \beta$) = .80, the available sample size ($N = 64$) allowed detection of effects of at least $f^2 = 0.13$. This again corresponds to approximately $r = .34$, indicating that the study was powered to detect medium effects, whereas smaller effects may not have been reliably detected. This approximation does not account for the multilevel structure of the data and should

therefore be interpreted cautiously. Moreover, given this sensitivity threshold, statistically significant effects close to this boundary may be subject to effect size inflation.”

We also added a new limitations section to the Discussion, where we address the reliability of these results:

“While the present findings provide initial insights into the role of causal attributions in the process of self-belief formation, several considerations are important for their interpretation. First, the sample size may limit statistical power and the robustness of the observed effects. A sensitivity analysis indicated that the present design was primarily powered to detect medium-sized associations, suggesting that smaller effects may not have been reliably detected. Accordingly, the observed associations between self-esteem and attributional biases should be interpreted with some caution and warrant further investigation in larger samples to establish their stability and generalizability.”

6. The authors note that the current sample shows predominantly subclinical symptom levels. Why is this the case? Was the sample selectively recruited? This characteristic may be related to the observed negativity bias and should be discussed.

Thank you for this comment. The use of the word “subclinical” was misleading. What we meant was that participants were not selectively recruited for clinical depression. The sample consisted of students, and we did not apply any exclusion criteria related to current or past depressive disorders, nor did we conduct clinical assessments or ask about diagnoses. The only measure of depressive symptoms was the PHQ-9, which can be used for diagnostic purposes, but does not replace a professional clinical assessment. While some participants scored above the typical cut-off for potential depression, this does not constitute a formal diagnosis. Thus, the sample reflects natural variability in depressive symptom levels in a student population, rather than selective recruitment. The reported negativity bias aligns with our previous work using the LOOP task, and we do not believe it is driven by the subclinical symptom levels in this sample.

In the Discussion section, we have included a paragraph on study limitations in which we now specify the nature of our sample:

“Second, given that the study was conducted in a student sample, depressive symptom levels were likely to fall predominantly below clinical thresholds, and participants’ diagnostic status was not assessed. As a result, effects involving depressive symptoms may be underestimated due to limited between-subject variability in PHQ-9 scores. Stronger or qualitatively different effects might emerge in samples that include individuals with a clinical diagnosis of depression.”

7. How was the sample size determined?

Sample size was determined a priori using a power analysis. We now describe this in full detail in Supplementary Note 2:

“An a priori power analysis was conducted using G*Power version 3.1.9.7 (Faul et al., 2007) to determine the required sample size. We initially planned to analyze four learning rates (one learning rate per experimental condition) in a general linear model with the two factors Interference (agent vs. no agent) x Ability (high ability vs low ability) and depression scores as a covariate. This design was approximated using an F-test for fixed effects (ANOVA: fixed effects, special, main effects and interactions). Assuming a medium to large effect size ($f = 0.35$), an alpha level of .05, and desired power of .80, the analysis indicated a required total sample size of $N = 67$. This calculation focused on detecting the interaction between Interference and Ability, which was the primary effect of interest.”

8. The statement that “two participants were excluded due to missing variance in their responses” requires clarification. What exactly is meant by “missing variance”?

By “missing variance,” we refer to participants whose expectation ratings showed little to no variability across trials. Specifically, these two participants did not adjust the response cursor on the majority of trials (i.e., in more than 60 trials), resulting in near-constant ratings (default value of 50). Visual inspection indicated that cursor movement ceased after only a few trials within each condition (see Figure R4 below). These response patterns suggest that the ratings did not reflect meaningful trial-by-trial engagement with the task and therefore limited the interpretability of their data. We replaced “missing variance” with “limited variability” in the main manuscript to make this clearer.

Figure R4: Trajectory of expectation ratings for excluded participants.

9. The manuscript states that trials from all conditions were intermixed in a fixed order. Why was a fixed order used, and how was this order determined?

Thank you for this question. There were several reasons why we chose a fixed, intermixed trial order rather than randomizing trials. First, using a fixed sequence ensured reproducibility and comparability across participants. This is important from a psychological perspective, as participants were exposed to comparable patterns of positive and negative “surprises” (prediction errors) as well as similar transitions between conditions.

Second, a fixed trial sequence provides consistent and comparable trial-by-trial learning signals, enabling more robust estimation of learning rate parameters across participants. In contrast, randomization would introduce additional variability in the sequence of prediction errors between subjects, which may substantially affect individual learning trajectories and parameter estimates even when the overall task structure is identical.

Third, because the LOOP task was also designed for potential use in an fMRI environment, a fixed order ensured consistency in temporal structure across participants, which is important for modeling task-related responses and maintaining comparability of neural signals across runs and individuals.

The sequence was generated using a pseudo-randomization procedure and validated in different samples (Czekalla et al., 2024; Müller-Pinzler et al., 2022; Schröder et al., 2025). Please note that these previous studies included an “Other” condition, as described in the response to comment #1. The trial sequence was retained from these studies to preserve its structure, with the only modification being that the “Other” condition trials were replaced by Interference (Agent) trials.

We now briefly justify the use of a fixed trial order in the Methods section:

“Trials from all conditions were intermixed in a fixed order, with no more than two consecutive trials of the same condition. This fixed sequence was used to ensure identical trial structure across participants, maintain a balanced distribution of conditions throughout the task, and provide reliable and comparable trial-by-trial signals for computational modeling, enabling robust estimation of learning rate parameters.”

10. The feedback was determined by a series of fixed prediction errors relative to participants' current beliefs. The exact procedure for generating and implementing these prediction errors needs to be described in detail.

Thank you for this helpful comment. We have clarified the procedure for generating and implementing prediction errors in a new paragraph within the Methods section:

“Instead of providing fixed feedback values, the feedback sequence was designed to elicit prediction errors of specific magnitudes and valences (Czekalla et al., 2024; Müller-Pinzler et al., 2022; Schröder et al., 2025). Planned prediction errors were drawn from a hand-designed sequence that followed a predefined distribution for each condition (High Ability: -18 to 27, 70% positive, 30% negative; Low Ability: -27 to 18, 30% positive, 70% negative). The sequence was identical for all participants and followed a fixed order within each experimental condition. For each trial, the feedback value was calculated by adding the corresponding planned prediction error from the sequence to the participant's current ability belief, defined as the average of

their last five expectation ratings within the respective condition. Before participants had provided their first performance expectation rating, the expectation value was set to 50%. This approach resulted in varying feedback sequences across participants while keeping actual prediction errors largely independent of individual performance expectations. It also ensured a relatively balanced distribution of actual negative prediction errors (Agent: $M = -12.7$, 45.9% of trials; No Agent: $M = -12.6$, 45.4%) and positive prediction errors (Agent: $M = 14.5$, 54.1%; No Agent: $M = 14.2$, 54.6%) across conditions.”

Notably, this prediction error sequence has been developed and validated in multiple independent samples and previous studies (Czekalla et al., 2024; Müller-Pinzler et al., 2022; Schröder et al., 2025). While the planned prediction errors were used to generate the feedback, the actual prediction errors experienced by participants could deviate from these values because participants’ trial-by-trial expectations did not always match the moving average used for feedback generation. Nevertheless, the predefined sequence ensured that the overall distribution of experienced prediction errors closely approximated the intended 70/30 ratio in each Ability condition. As noted above, this procedure also led to relatively balanced distributions of positive and negative prediction errors in both Interference conditions.

11. The influence of relative probability density on feedback values was modeled as linear. Have alternative, potentially nonlinear mappings been considered?

Thank you for this question. This is something that we carefully considered during the design of the modeling approach. First, I would like to clarify in more detail how the relative probability density of the percentile feedback is structured. The Figure R5 below (adapted from our previous publication, Müller-Pinzler et al., 2022) illustrates the relative probability density associated with different feedback values. As shown in the figure, the distribution follows a nonlinear pattern with a “normal” decay towards the ends of the feedback scale. In our previous publications we also considered an alternative implementation using a linear decay for feedback values deviating from the midpoint (50; see Figure R5 left side). However, from a theoretical perspective it may appear more plausible to assume a distribution that reflects how many types of performance data encountered in everyday life are typically distributed – namely according to a normal distribution. Because many real-life performance outcomes are approximately normally distributed, extreme values occur less frequently than values near the center of the distribution. Through everyday experience with such performance distributions, individuals may therefore implicitly expect extreme feedback values to occur less frequently than values near the center of the scale. The nonlinear relative probability density implemented in our model reflects this assumption. Of course, we cannot be certain whether participants implicitly represent the feedback structure in this exact way. However, model comparisons in our previous studies consistently supported this assumption. In those analyses, the model including the nonlinear “normal” decay outperformed both models without a decay factor and models implementing a linear decay.

Based on both considerations – the theoretical idea of the relative probability density and our previous empirical experience with model comparisons – we opted for a more parsimonious modeling approach in the present study and therefore did not include the linear decay model.

Figure R5. Depiction of the linear decay (left) and the decay following the relative probability density of the normal distribution (right) for the different feedback values. The figure is adapted from Müller-Pinzler et al. 2022.

12. Typo: “On average, participants showed higher learning rates for negative (Mdn = .024)” should be Mdn = .24.

Thank you, we corrected this.

References

- Bandura, A. (1986). *Social foundations of thought and action*. Prentice Hall.
- Bandura, A. (1997). *Self-efficacy: The exercise of control*. W.H. Freeman.
- Brotzeller, F., & Gollwitzer, M. (2025). Exploring asymmetries in self-concept change after discrepant feedback. *Personality & Social Psychology Bulletin*, 51(9), 1731–1744.
- Campbell, W. K., & Sedikides, C. (1999). Self-threat magnifies the self-serving bias: A meta-analytic integration. *Review of General Psychology: Journal of Division 1, of the American Psychological Association*, 3(1), 23–43.
- Czekalla, N., Schröder, A., Mayer, A. V., Stierand, J., Stolz, D. S., Kube, T., Korn, C. W., Wilhelm-Groch, I., Klein, J. P., Paulus, F. M., Krach, S., & Müller-Pinzler, L. (2024). Aberrant insula activity to negative and reduced learning from positive prediction errors as mechanisms underlying maladaptive self-belief formation in depression. In *bioRxiv* (p. 2024.05.09.593087). <https://doi.org/10.1101/2024.05.09.593087>
- Dorfman, H. M., Bhui, R., Hughes, B. L., & Gershman, S. J. (2019). Causal inference about good and bad outcomes. *Psychological Science*, 30(4), 516–525.
- Dorfman, H. M., Tomov, M. S., Cheung, B., Clarke, D., Gershman, S. J., & Hughes, B. L. (2021). Causal inference gates corticostriatal learning. *The Journal of Neuroscience: The Official Journal of the Society for Neuroscience*, 41(32), 6892–6904.
- Elig, T. W., & Frieze, I. H. (1979). Measuring causal attributions for success and failure. *Journal of Personality and Social Psychology*, 37(4), 621–634.
- Faul, F., Erdfelder, E., Lang, A.-G., & Buchner, A. (2007). G*Power 3: a flexible statistical power analysis program for the social, behavioral, and biomedical sciences. *Behavior Research Methods*, 39(2), 175–191.
- Garrett, N., Sharot, T., Faulkner, P., Korn, C. W., Roiser, J. P., & Dolan, R. J. (2014). Losing the rose tinted glasses: neural substrates of unbiased belief updating in depression. *Frontiers in Human Neuroscience*, 8, 639.

- Hoffmann, J. A., Hobbs, C., Moutoussis, M., & Button, K. S. (2024). Lack of optimistic bias during social evaluation learning reflects reduced positive self-beliefs in depression and social anxiety, but via distinct mechanisms. *Scientific Reports*, *14*(1), 22471.
- Korn, C. W., Sharot, T., Walter, H., Heekeren, H. R., & Dolan, R. J. (2014). Depression is related to an absence of optimistically biased belief updating about future life events. *Psychological Medicine*, *44*(3), 579–592.
- Malle, B. F. (2006). The actor-observer asymmetry in attribution: a (surprising) meta-analysis. *Psychological Bulletin*, *132*(6), 895–919.
- Marsh, H. W. (1990). A multidimensional, hierarchical model of self-concept: Theoretical and empirical justification. *Educational Psychology Review*, *2*(2), 77–172.
- Mezulis, A. H., Abramson, L. Y., Hyde, J. S., & Hankin, B. L. (2004). Is there a universal positivity bias in attributions? A meta-analytic review of individual, developmental, and cultural differences in the self-serving attributional bias. *Psychological Bulletin*, *130*(5), 711–747.
- Müller-Pinzler, L., Czekalla, N., Mayer, A. V., Schröder, A., Stolz, D. S., Paulus, F. M., & Krach, S. (2022). Neurocomputational mechanisms of affected beliefs. *Communications Biology*, *5*(1), 1241.
- Müller-Pinzler, L., Czekalla, N., Mayer, A. V., Stolz, D. S., Gazzola, V., Keysers, C., Paulus, F. M., & Krach, S. (2019). Negativity-bias in forming beliefs about own abilities. *Scientific Reports*, *9*(1), 14416.
- Schröder, A., Czekalla, N., Mayer, A. V., Zhang, L., Stolz, D. S., Korn, C. W., Diekelmann, S., Luebber, F., Paulus, F. M., Müller-Pinzler, L., & Krach, S. (2025). Initial expectations and confidence affect the formation of novel self-beliefs and their revision. *Open Mind : Discoveries in Cognitive Science*, *9*, 1576–1596.
- Sharot, T. (2011). The optimism bias. *Current Biology*, *21*(23), R941–R945.
- Silver, W. S., Mitchell, T. R., & Gist, M. E. (1995). Responses to successful and unsuccessful performance: The moderating effect of self-efficacy on the relationship between performance and attributions. *Organizational Behavior and Human Decision*

Processes, 62(3), 286–299.

Stajkovic, A. D., & Sommer, S. M. (2000). Self- efficacy and causal attributions: Direct and reciprocal links. *Journal of Applied Social Psychology*, 30(4), 707–737.

Zamfir, E., & Dayan, P. (2022). Interactions between attributions and beliefs at trial-by-trial level: Evidence from a novel computer game task. *PLoS Computational Biology*, 18(9), e1009920.

Zuckerman, M. (1979). Attribution of success and failure revisited, or: The motivational bias is alive and well in attribution theory. *Journal of Personality*, 47(2), 245–287.

Causal attributions shape the formation of novel ability self-beliefs

Mayer, Schröder, et al.

Response to reviewers, second revision

Reviewer #1

1. Although the authors have softened some of the causal language and added a useful limitations section, I still think that the overall scope of the Discussion and Conclusion remains somewhat broader than the present data can directly support. As currently analyzed, the study most directly shows a negativity bias in self-related learning; associations of learning bias scores with depressive symptoms and self-esteem; and an effect of prediction error valence on external attributions in both attribution models. In addition, depressive symptoms were associated with lower external attributions overall, whereas self-esteem showed a significant interaction with prediction error valence. However, the manuscript continues to move toward broader claims about “self-belief formation,” “maintenance” of self-beliefs, and “enduring negative self-beliefs”. In particular, “formation” does not appear to have been operationalized as a distinct outcome beyond belief updating, and “maintenance” would seem to require stronger longitudinal or recursive evidence.

Thank you for this remark regarding the terms “formation” and “maintenance”. We have now softened the language and speak more precisely about “self-ability belief formation” or “formation of self-ability beliefs”. We have deleted the term “maintenance” and agree that in the current work, this term would be too broad and not justified. However, we would like to insist on keeping the term “formation” (concatenated trials, allowing to establish a new higher-order ability self-belief) to describe what is novel about the current task design and is different from “belief updating” (isolated trials, not connected on a higher-order level). In a recent preprint, currently in the re-submission process, we explicitly outline why we refer to “formation” using the LOOP task and what distinguishes the task from classic belief-updating tasks. We have also cited the respective preprint (Krach et al., 2024). As this distinction between updating and formation is not at the core of the present manuscript, we simply refer to the preprint in the introduction:

“Addressing these gaps is essential for understanding how moment-to-moment causal interpretations contribute to the formation of self-related ability beliefs (Krach et al., 2024).”

Given that the sample consists of undiagnosed students and that the authors themselves acknowledge the lack of directional or causal inference, I would encourage a further narrowing of the mechanistic and clinical claims so that they more closely match the actual evidential scope of the study. Relatedly, the final sentence of the abstract may still overstate the scope of the findings. The sample consists of individuals from a non-clinical student population with varying levels of depressive symptoms, rather than patients with a clinical diagnosis of depression. Formulations referring directly to effects “in depression” therefore risk exceeding what the present data can support. A more cautious formulation referring to depressive symptom severity or tendency in a non-clinical sample would be more appropriate.

We agree that a more cautious interpretation and wording are appropriate in this regard. The final sentence of the abstract now reads as follows: “These findings provide insight into cognitive mechanisms associated with negative self-related ability beliefs and may help to inform our understanding of processes linked to depressive symptoms.”

2. The term “self-belief” may be too broad for the construct actually measured in this study. Based on the task and outcome measures, the study appears to assess updating of performance- or ability-related expectations, rather than self-beliefs in a broader sense. More specific language, such as “self-related belief updating,” “beliefs about one’s own ability,” or “performance-related expectations/confidence,” may better reflect the construct actually assessed. The authors may also find it useful to consider recent work distinguishing confidence from broader self-beliefs (Hoven, M., Luijckes, J., Denys, D. et al. How do confidence and self-beliefs relate in psychopathology: a transdiagnostic approach. *Nat. Mental Health* 1, 337–345 (2023). <https://doi.org/10.1038/s44220-023-00062-8>).

Similar to our first commentary, we agree that the term “self-belief” may include other facets than simply beliefs related to ability (e.g., personality, appearance, health etc.). We have thus corrected this in the entire manuscript and now speak of “self-related ability-beliefs”, “beliefs about one’s own ability”, or “ability self-beliefs” instead.

3. It is also unclear why self-esteem is not discussed alongside depression in the third hypotheses section (We also anticipated that with increasing depressive symptom severity individuals would more likely attribute failures, defined as worse-than-expected feedback, to internal causes, while attributing successes, defined as better-than-expected feedback, to external causes, as implicated by the attributional theory of depression), given that both constructs are examined throughout the manuscript.

We agree with the reviewer and added a sentence as follows: “As implicated by the attributional theory of depression (Abramson et al., 1978; Fresco et al., 2006; Heimberg et al., 1989; Huang, 2015; Southall & Roberts, 2002), we also anticipated that individuals with more severe depressive symptoms and **lower self-esteem** would be more likely to attribute failures (worse-than-expected feedback) to internal causes and successes (better-than-expected feedback) to external causes, reflecting a reduced or even reversed self-serving attributional bias.”

4. It is also notable that the concept of “self-serving bias” is introduced for the first time in the Results section rather than in the Introduction.

Along with this suggestion, we introduced the term “self-serving bias” in the revised introduction. The two adapted paragraphs now read as follows:
“For instance, while healthy individuals tend to attribute successes to internal causes (e.g., their abilities) and failures to external causes (e.g., bad luck) (Larson, 1977), a pattern commonly referred to as a **self-serving attributional bias** or self-serving attributional style, individuals with depression often show the opposite pattern (Kuiper, 1978; Peterson & Seligman, 1984; Sweeney et al., 1986).”

“As implicated by the attributional theory of depression (Abramson et al., 1978; Fresco et al., 2006; Heimberg et al., 1989; Huang, 2015; Southall & Roberts, 2002), we also anticipated

that individuals with more severe depressive symptoms and lower self-esteem would be more likely to attribute failures (worse-than-expected feedback) to internal causes and successes (better-than-expected feedback) to external causes, reflecting a reduced or even reversed **self-serving attributional bias**.”

5. Where interactions are reported as significant, it would be helpful to provide more explicit follow-up analyses or a clearer explanation of their direction and form. At present, some interaction effects are noted as significant, but the substantive pattern they represent remains somewhat underexplained.

We have now added a more extensive description of significant interaction effects specifically when reporting results of the model-agnostic analysis:

“To further decompose the three-way interaction, we examined expectation change across trials separately for each condition (Figure 1C). In the No Agent condition, participants showed pronounced updating, with expectations increasing over time in the High Ability condition ($\beta = 0.10 [-0.09, 0.28]$, $p = .301$) and decreasing in the Low Ability condition ($\beta = -0.47 [-0.65, -0.29]$, $p < .001$). In contrast, in the Agent condition, expectations decreased in both Ability conditions, though effects were weaker and only marginal in the High Ability condition (Low Ability: $\beta = -0.23$, $SE = 0.09$, $95\% CI [-0.41, -0.05]$, $p = .014$; High Ability: $\beta = -0.18$, $SE = 0.09$, $95\% CI [-0.36, 0.00]$, $p = .052$). Follow-up comparisons showed a significant difference between High and Low Ability in the No-Agent condition ($\Delta\beta = -0.57$, $95\% CI [-0.72, -0.41]$, $p < .001$), but not in the Agent condition ($\Delta\beta = -0.05$, $95\% CI [-0.20, 0.10]$, $p = .536$), indicating that feedback-dependent differences in updating were attenuated when outcomes could be attributed to an external agent.”

6. The Discussion does acknowledge that depressive symptoms and self-esteem yielded different attributional results, which I appreciate. I also value the authors’ attempt to explain this asymmetry. However, the current explanation could be strengthened by a more explicit theoretical account, for example, by clarifying the conceptual and mechanistic differences between depression and self-esteem. These patterns may instead reflect different psychological processes and carry different theoretical implications. Developing this point further would, in my view, substantially strengthen the theoretical contribution of the Discussion.

We agree that the asymmetrical findings for depressive symptoms and self-esteem warrant a clearer theoretical interpretation. In response, we expanded the Discussion to provide a more explicit theoretical account of the differential associations of depressive symptoms and self-esteem with attributional bias:

“More importantly, depressive symptoms and self-esteem may reflect partly distinct psychological processes with different relevance for attributional style. Depression, as assessed using the PHQ-9, captures a broad and heterogeneous constellation of affective, cognitive, and somatic symptoms (Kroenke et al., 2001), many of which may not be directly linked to causal attributions of success and failure. By contrast, self-esteem reflects global self-evaluative beliefs (Hoven et al., 2023) and may therefore be conceptually more proximal to the ability-related self-beliefs examined in the present study (Duval & Silvia, 2002; Fitch, 1970; Romney, 1994; Shepperd et al., 2008). From this perspective, the reduced self-serving

bias commonly described in depression (Kuiper, 1978; Peterson & Seligman, 1984; Sweeney et al., 1986) may be more strongly related to low self-esteem than to depressive symptom severity per se (Tennen et al., 1987) and may therefore also extend to other forms of psychopathology characterized by diminished self-esteem, such as anxiety disorders (Maldonado et al., 2013; Sowislo & Orth, 2013). Future studies comparing depressive and non-depressive clinical populations may help disentangle the relative contributions of depressive symptoms and self-esteem to attributional biases.”

7. Phrases such as “distinct attributional patterns and biased learning” may overstate what the present attribution analyses can support, because depressive symptoms and self-esteem were modeled separately and not directly compared. As such, the current results do not seem sufficient to conclude that they reflect distinct attributional patterns.

We agree that this was misleading and could be read as if we expect different or even opposing effects for depressive symptom severity and self-esteem. We therefore rephrased this sentence: “Specifically, we assessed whether attributions of feedback in a validated learning task influence belief updating, and whether individual differences in self-esteem and depressive symptoms are associated with systematic biases in attributions and learning.”

8. The phrasing “with increasing depressive symptom severity individuals” can be more clearly formulated, such as “individuals with more severe depressive symptoms” or “greater depressive symptom severity was associated with....”

We agree with the reviewer and changed the phrasing as follows: “As implicated by the attributional theory of depression (Abramson et al., 1978; Fresco et al., 2006; Heimberg et al., 1989; Huang, 2015; Southall & Roberts, 2002), we also anticipated that **individuals with more severe depressive symptoms and lower self-esteem** would be more likely to attribute failures (worse-than-expected feedback) to internal causes and successes (better-than-expected feedback) to external causes, reflecting a reduced or even reversed self-serving attributional bias.”

References

- Abramson, L. Y., Seligman, M. E., & Teasdale, J. D. (1978). Learned helplessness in humans: Critique and reformulation. *Journal of Abnormal Psychology, 87*(1), 49–74.
- Duval, T. S., & Silvia, P. J. (2002). Self-awareness, probability of improvement, and the self-serving bias. *Journal of Personality and Social Psychology, 82*(1), 49–61.
- Fitch, G. (1970). Effects of self-esteem, perceived performance, and choice on causal attributions. *Journal of Personality and Social Psychology, 16*(2), 311–315.
- Fresco, D. M., Alloy, L. B., & Reilly-Harrington, N. (2006). Association of attributional style for negative and positive events and the occurrence of life events with depression and anxiety. *Journal of Social and Clinical Psychology, 25*(10), 1140–1160.
- Heimberg, R. G., Klosko, J. S., Dodge, C. S., Shadick, R., Becker, R. E., & Barlow, D. H. (1989). Anxiety disorders, depression, and attributional style: A further test of the specificity of depressive attributions. *Cognitive Therapy and Research, 13*(1), 21–36.
- Hoven, M., Luigjes, J., Denys, D., Rouault, M., & van Holst, R. J. (2023). How do confidence and self-beliefs relate in psychopathology: a transdiagnostic approach. *Nature Mental Health, 1*(5), 337–345.
- Huang, C. (2015). Relation between attributional style and subsequent depressive symptoms: A systematic review and meta-analysis of longitudinal studies. *Cognitive Therapy and Research, 39*(6), 721–735.
- Krach, S., Müller-Pinzler, L., Czekalla, N., Schröder, A., Wilhelm-Groch, I., Luebber, F., Rademacher, L., Stolz, D., Paulus, F. M., & Mayer, A. V. (2024). *Self-belief formation*. <https://osf.io/preprints/psyarxiv/2y5tv>
- Kroenke, K., Spitzer, R. L., & Williams, J. B. (2001). The PHQ-9: validity of a brief depression severity measure. *Journal of General Internal Medicine, 16*(9), 606–613.
- Kuiper, N. A. (1978). Depression and causal attributions for success and failure. *Journal of Personality and Social Psychology, 36*(3), 236–246.
- Larson, J. R., Jr. (1977). Evidence for a self-serving bias in the attribution of causality.

Journal of Personality, 45(3), 430–441.

Maldonado, L., Huang, Y., Chen, R., Kasen, S., Cohen, P., & Chen, H. (2013). Impact of early adolescent anxiety disorders on self-esteem development from adolescence to young adulthood. *The Journal of Adolescent Health: Official Publication of the Society for Adolescent Medicine*, 53(2), 287–292.

Peterson, C., & Seligman, M. E. (1984). Causal explanations as a risk factor for depression: Theory and evidence. *Psychological Review*, 91(3), 347–374.

Romney, D. M. (1994). Cross-validating a causal model relating attributional style, self-esteem, and depression: an heuristic study. *Psychological Reports*, 74(1), 203–207.

Shepperd, J., Malone, W., & Sweeny, K. (2008). Exploring causes of the self-serving bias: The self-serving bias. *Social and Personality Psychology Compass*, 2(2), 895–908.

Southall, D., & Roberts, J. E. (2002). Attributional Style and Self-Esteem in Vulnerability to Adolescent Depressive Symptoms Following Life Stress: A 14-Week Prospective Study. *Cognitive Therapy and Research*, 26(5), 563–579.

Sowislo, J. F., & Orth, U. (2013). Does low self-esteem predict depression and anxiety? A meta-analysis of longitudinal studies. *Psychological Bulletin*, 139(1), 213–240.

Sweeney, P. D., Anderson, K., & Bailey, S. (1986). Attributional style in depression: A meta-analytic review. *Journal of Personality and Social Psychology*, 50(5), 974–991.

Tennen, H., Herzberger, S., & Nelson, H. F. (1987). Depressive attributional style: the role of self-esteem. *Journal of Personality*, 55(4), 631–660.